# Bladder-cancer-associated mutations in *RXRA* activate peroxisome proliferator-activated receptors to drive urothelial proliferation

Angela M Halstead[1], Chiraag D Kapadia[1], Jennifer Bolzenius[1], Clarence E Chu[1], Andrew Schriefer[2], Lukas D Wartman[1], Gregory R Bowman[3], Vivek K Arora[1]*

[1]Department of Internal Medicine, Division of Oncology, Washington University School of Medicine, St Louis, United States; [2]Genome Technology Access Center, Washington University School of Medicine, St Louis, United States; [3]Department of Biochemistry and Molecular Biophysics, Washington University School of Medicine, St Louis, United States

**Abstract** RXRA regulates transcription as part of a heterodimer with 14 other nuclear receptors, including the peroxisome proliferator-activated receptors (PPARs). Analysis from TCGA raised the possibility that hyperactive PPAR signaling, either due to PPAR gamma gene amplification or RXRA hot-spot mutation (S427F/Y) drives 20–25% of human bladder cancers. Here, we characterize mutant RXRA, demonstrating it induces enhancer/promoter activity in the context of RXRA/PPAR heterodimers in human bladder cancer cells. Structure-function studies indicate that the RXRA substitution allosterically regulates the PPAR AF2 domain via an aromatic interaction with the terminal tyrosine found in PPARs. In mouse urothelial organoids, PPAR agonism is sufficient to drive growth-factor-independent growth in the context of concurrent tumor suppressor loss. Similarly, mutant RXRA stimulates growth-factor-independent growth of *Trp53/Kdm6a* null bladder organoids. Mutant RXRA-driven growth of urothelium is reversible by PPAR inhibition, supporting PPARs as targetable drivers of bladder cancer.
DOI: https://doi.org/10.7554/eLife.30862.001

*For correspondence:
arorav@wustl.edu

## Introduction

Bladder cancer is the sixth most common cancer in the US and is predicted to cause ~17,000 deaths in 2017 [The website of the National Cancer Institute (https://www.cancer.gov)]. In contrast to other common malignancies, bladder cancer management has not yet benefited from molecularly targeted therapies. Early phase clinical data with agents targeting FGFR3 or ERBB family members have shown encouraging activity in the minority of patients with oncogenic versions of these kinases (*Nogova et al., 2017*; *Choudhury et al., 2016*), but identification of new targetable genomic drivers is needed to expand the armamentarium. Nuclear receptors are amongst the most successful therapeutic targets in oncology, with small molecule inhibitors of estrogen receptor (ER) and androgen receptor (AR) improving survival in breast and prostate cancers (*Fisher et al., 1996*; *Beer et al., 2014*). The established ability to modulate nuclear receptors with drugs makes them an attractive class of targets for other cancer types when there is compelling evidence supporting their role as tumor drivers.

RXRA is a nuclear receptor that regulates transcription as a homodimer or as an obligate heterodimerization partner for 14 other nuclear receptors, including the three peroxisome proliferator-activated receptors PPARA, PPARD, and PPARG (*Evans and Mangelsdorf, 2014*). Exome analysis of

**eLife digest** Bladder cancer is the sixth most common type of cancer in the United States. At the moment, treatment options for advanced bladder cancer are limited to chemotherapy and immunotherapy, both of which benefit only some patients. Many other types of cancer can be treated with drugs that are specific to genetic mutations found in those cancer cells, often making the treatments more efficient with fewer side effects.

Between 5–8% of people with bladder cancer have a mutation in the gene that produces a protein called RXRA. This protein partners with itself or with other proteins to control gene activity. However, it was not clear what mutant RXRA proteins do in bladder cancer cells.

Halstead et al. studied the RXRA mutation in human bladder cancer cells and "mini-bladders" grown in the laboratory from mouse bladder cells. Biochemical experiments showed that the mutant RXRA protein causes abnormally high activity in one group of its partner proteins, called peroxisome proliferator-activated receptors (PPARs). The PPARs, in turn, switch on genes that help cancer cells to grow and multiply. Computational simulations of the mutant RXRA binding to PPARs revealed, at a molecular level, how this activation occurs. Lastly, Halstead et al. used chemicals that block the activity of PPARs to stop the growth of cells in the mouse mini-bladders that contained the RXRA mutation.

These findings suggest that bladder cancer patients with the RXRA mutation may benefit from therapies that inhibit PPARs. Such therapies could also benefit the approximately 15–20% of people with bladder cancer who do not have the RXRA mutation but who do have over-active PPARs. Although there are chemicals that block the activity of PPARs, more research is needed to refine them before they can be used to treat cancer.

DOI: https://doi.org/10.7554/eLife.30862.002

bladder cancer samples across three independent cohorts identified mutations at RXRA S427 in 5–8% of cases, always leading to an amino acid substitution with an aromatic amino acid, phenylalanine (~5%) or tyrosine (~1%) (*Cancer Genome Atlas Research Network, 2014*; *Guo et al., 2013*; *Van Allen et al., 2014*). Reported analysis from TCGA found evidence of up-regulated PPAR pathway activity in RXRA hot-spot mutant cases (*Cancer Genome Atlas Research Network, 2014*). Copy number analysis identified PPARG as amplified in 17% of bladder cancer cases (*Cancer Genome Atlas Research Network, 2014*), with PPARG amplified cases being highly enriched for PPARG mRNA expression (q value = 2.6 × 10-9) (*Cerami et al., 2012*). These data raised the possibility that hyperactive PPAR signaling, either due to gene-amplification-driven hyper-expression or RXRA hot-spot mutation may drive 20–25% of bladder cancers. Here, we characterize structural and biological consequences of the RXRA hot-spot mutations.

## Results

### Mutant RXRA induces PPAR transcriptional signaling at the canonical DR1 motif

To establish a causal role of RXRA hot-spot mutations in hyper-activation of PPAR singling, we used retroviral transduction to introduce wild-type or mutant RXRA into two bladder cancer cell lines, JMSU-1 and 575A. Equivalent RXRA expression levels were confirmed by qPCR and western blot (*Figure 1A*). RNA-seq was used to compare transcriptomes in the wild-type and S427F RXRA expressing cells. Gene transcripts robustly up-regulated by the mutation (twofold, FDR <0.05) were analyzed using over-representation analysis (ORA) to identify enriched pathways. In both cell lines, the PPAR signaling pathway (KEGG-hsa03320) was the top scoring hit. When the ORA was limited to genes up-regulated by the mutation in both cell lines, again PPAR signaling was the top hit (*Figure 1B*, *Figure 1—source data 1*). We also compared transcriptome changes induced by the RXRA mutation to those induced after treating RXRA$^{wt}$ expressing cells with the PPARG agonist pioglitazone. There was robust correlation in both JMSU-1 and 575A (Pearson $r$ = 0.72, p=1.4 × 10$^{-34}$ or 3.1 × 10$^{-105}$), confirming that the mutation drives expression changes similar to agonist-induced PPAR activation (*Figure 1C*). Mutation-driven hyper-activation of *PLIN2* and *FABP3*, two genes

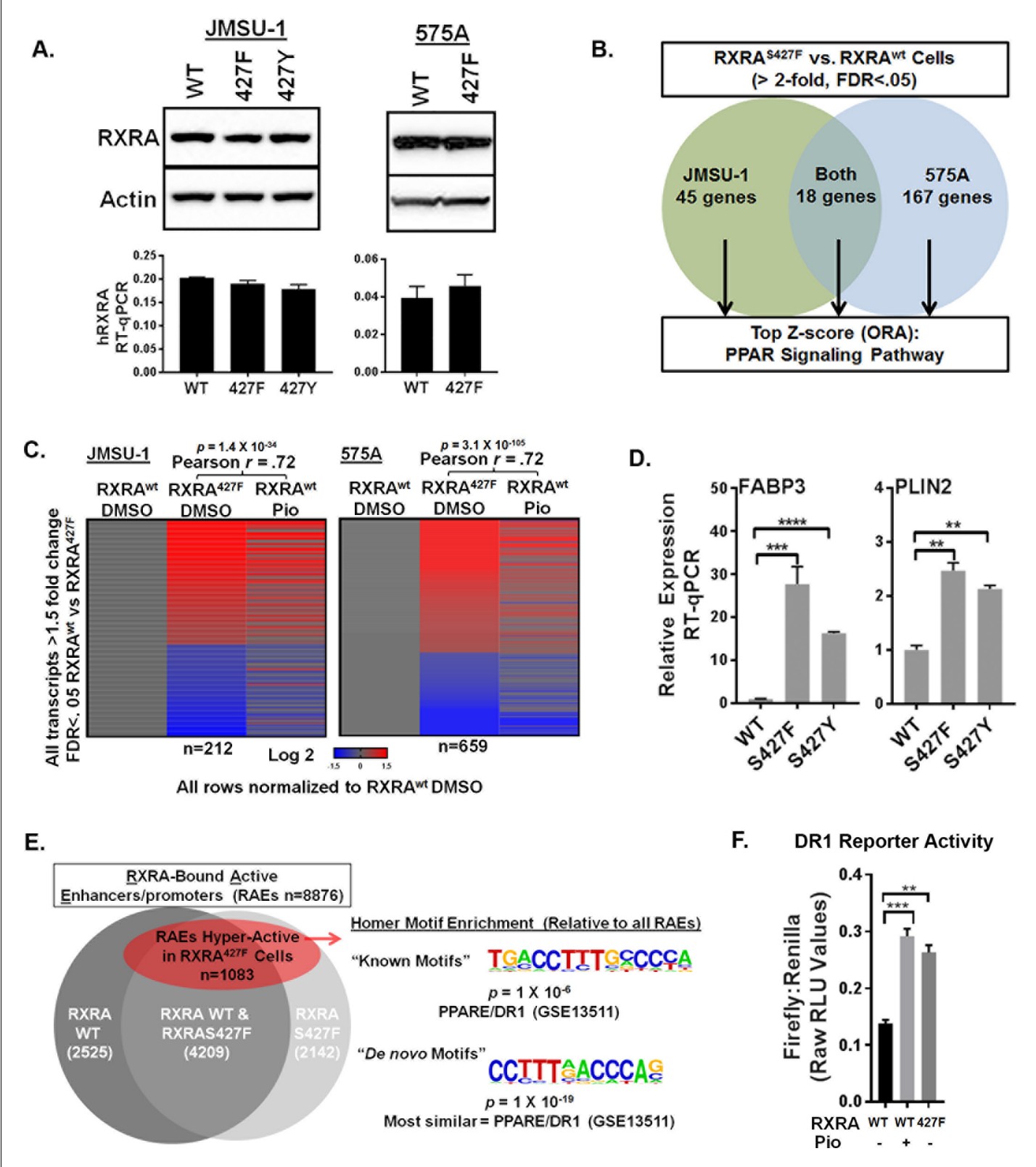

**Figure 1.** RXRA hot-spot mutations induce the PPAR signaling pathway by activating enhancer/promoters with a canonical PPAR response element. (**A**) JMSU-1 and 575A cells were transduced with pBABE retrovirus to express indicated RXRA alleles and expression confirmed by western blot (top) or RT-qPCR in triplicate ± SD (data expressed as a fraction of actin signal). (**B**) Protein coding transcripts up-regulated greater than or equal to twofold

*Figure 1 continued on next page*

*Figure 1 continued*

(FDR < 0.05) in cells expressing RXRA$^{S427F}$ compared to cells expressing RXRA$^{wt}$ were identified and then subjected to over representation analysis (ORA, GO-Elite) to discover enriched pathways relative to all other protein coding transcripts identified by RNA-seq. Experiment was done in two bladder cancer cell lines, JMSU-1 or 575A, using three RNA samples, each purified from an independent cell well, for each condition. (See also source data 1). (C) Transcriptome changes induced by RXRA$^{S427F}$ relative to RXRA$^{wt}$ were compared to expression changes of the same transcripts induced by 16 hr of pioglitazone (1 μM) treatment in the RXRA$^{wt}$ expressing cells. D) Relative expression of two PPAR targets with expression of indicated *RXRA* alleles. RT-qPCR performed in triplicate ±SD. Comparison by Student's t-test. (E) RAEs were defined by the presence of overlapping ChIP-seq signal for RXRA and H3K27ac. RAEs identified by binding of RXRA$^{wt}$ and/or RXRA$^{S427F}$ are represented in grey. Hyperactive RAEs represented in red had elevated H3K27ac mean peak height in the mutant expressing cells compared to the wild-type cells (FDR < 0.05). All ChIP-seq peak callings were based on data from three independent immuno-precipitations, each utilizing input material from an independent cell plate. HOMER motif analysis was used to identify motifs enriched in hyperactive RAEs relative to the background of non-hyperactive RAEs. Source data 2 specifies number of peaks in each sector of the venn diagram. (F) Activity of a DR1 response element reporter (3X PPRE) transfected into JMSU-1 cells stably expressing either RXRA$^{wt}$ or RXRA$^{S427F}$. RXRA$^{wt}$ cells were also treated with pioglitazone (1 μM) for 16 hr. For all reporter assays, Firefly luciferase expressing reporter was co-transfected with a constitutive Renilla luciferase expression vector to normalize for transfection efficiency. Data represents mean ± SEM of Firefly to Renilla luciferase signal from three independent experiments done on different days, each performed using triplicate cell wells. Statistical comparisons are by paired *t*-test.

DOI: https://doi.org/10.7554/eLife.30862.003

The following source data is available for figure 1:

**Source data 1.** GO Elite Over Representation Analysis complete results.
DOI: https://doi.org/10.7554/eLife.30862.004

**Source data 2.** ChIP-seq peak numbers used to generate venn diagram in *Figure 1E*.
DOI: https://doi.org/10.7554/eLife.30862.005

found up-regulated in our RNA-seq analysis, also occured in the context of RXRA$^{S427Y}$ in JMSU-1 cells (*Figure 1D*), confirming gain-of-function with either aromatic substitution. Previous analysis of human bladder cancer specimens identified a correlation between the presence of RXRA$^{S427F/Y}$ and both the up-regulation of *PLIN2* expression and PPAR signaling pathway activity (*Cancer Genome Atlas Research Network, 2014*). Our data establishes a causal role of the RXRA mutations in driving these expression changes.

Canonical RXRA-mediated gene regulation involves direct engagement of promoter/enhancers of target genes in the context of homodimers or heterodimers. Specific partner-pairs preferentially bind to direct repeats (DR) with defined spacer lengths. RXR homodimers and RXR/PPAR heterodimers both show preference for DRs with a single nucleotide spacer (DR1) (*Evans and Mangelsdorf, 2014*; *Nielsen et al., 2008*). Using ChIP-seq data for RXRA and acetylated H3K27 (H3K27ac), we identified all active enhancer/promoters bound by RXRA$^{wt}$ and/or RXRA$^{S427F}$ in JMSU-1 ('RXR-bound Active Enhancer/promoters' or RAEs). Differential mean peak height of the H3K27ac signal was used as an indicator of relative activation status in the wild-type and mutant condition at each RAE. ~12% of RAEs showed significant up-regulation in the mutant condition (FDR <0.05) (*Figure 1E*, *Figure 1—source data 2*). HOMER motif finding was then performed to identify motifs enriched at these hyper-activated RAEs, relative to the background of all RAEs. Consistent with activation of PPAR pathway genes, the canonical RXR/PPAR (DR1) motif was the only known motif found to be enriched with significant q-value. The most statistically robust motif identified by de novo discovery is shown in *Figure 1E*. Comparing the de novo identified motif to known motifs found it to be most similar to a PPAR response element (DR1). These data demonstrate that hyper-activation of enhancer/promoters driven by the RXRA hot-spot mutation occurs preferentially at canonical RXR/PPAR response elements. Using transient transfection of a DR1-driven luciferase reporter, we confirmed relative hyper-activation of DR1 in the JMSU-1 cells expressing mutant RXRA compared to wild-type (*Figure 1F*), with a degree of induction similar to that observed after pioglitazone treatment.

## Hot-spot mutation-driven transcriptional activity is dependent on expression of PPARD or PPARG

Having established that mutant RXRA up-regulates PPAR target gene expression, we next tested if this was dependent on PPARs. TCGA data were first queried to determine which of the three PPARs are most highly expressed in bladder cancer specimens with mutant RXRA (*Cerami et al., 2012*).

Both *PPARG* and *PPARD* were significantly more highly expressed than *PPARA*, but the relative proportion of *PPARG* and *PPARD* varied amongst the samples (*Figure 2A,B*). Using siRNA, we acutely knocked-down PPARG and PPARD in the JMSU-1 and 575A cells transduced with wild-type or mutant RXRA and used RT-qPCR to query the expression of *PLIN2* and *FABP3* (*Figure 2C*). In the non-targeting control, both genes were again found up-regulated by mutant RXRA. PPARD knock-down alone appeared to increase PPARG expression suggesting receptor cross talk and loss of negative feedback regulation. However, expression of neither PPAR target gene was significantly reduced by PPARD knock-down alone. PPARG knock-down partially inhibited *PLIN2* expression in both cell lines, but did not have significant impact on *FABP3*. Combined knock-down of PPARD and PPARG, however, strongly inhibited RXRA$^{S427F}$-driven hyper-expression of both genes, to an extent greater than knockdown of PPARG alone. Thus, both PPARD and PPARG contribute to mutant RXRA-mediated transcriptional hyperactivity in human bladder cancer cells and appear to have redundant function.

Since hyperactivity of mutant RXRA was dependent on PPAR expression, we reasoned that no hyperactivity would be observed in the DR-1 luciferase reporter assay in a bladder cancer cell line with low endogenous expression of PPARs. Treatment of UM-UC-3 with agonists to either PPARG or PPARD did not have meaningful impact on reporter activity when RXRA was transfected without a PPAR (*Figure 2D*), indicating a lack of relevant endogenous expression in this cell line. Similarly, RXRA$^{S427F/Y}$ did not show increased activity relative to RXRA$^{wt}$ (*Figure 2D*). Since RXRA can drive transcription as a homodimer with a preference for the DR1 motif, these data indicate that the mutation does not confer gain-of-function in the homodimer context. We then examined reporter activity with co-transfection of PPARG or PPARD. Expression of either PPAR was sufficient to elicit mutant RXRA associated hyperactivity to an extent that approximated agonist induced activation of the PPAR in the wild-type RXRA condition (*Figure 2D*). Similar to what was observed in the JMSU-1 stable cells, RXRA$^{S427F}$ appeared to have somewhat stronger activity than RXRA$^{S427Y}$, but both were hyperactive relative to RXRA$^{wt}$ (*Figure 2D*). Lastly, we tested reporter activity with co-transfection of retinoic acid receptor alpha (RARA), which belongs to another family of RXRA heterodimerization partners. Both the DR1 reporter and a reporter with the preferred RXR/RAR binding motif, DR5, were induced by the RAR agonist all-trans-retinoic acid, but mutant RXRA did not elicit greater activity than wild-type RXRA (*Figure 2—figure supplement 1*). These data confirmed that RXRA hotspot mutations confer gain-of-function selectively in the context of PPAR expression.

## Mutant RXRA activates the PPAR AF2 via an allosteric mechanism

Nuclear receptors share a conserved domain structure and are prototypically activated by agonist binding to the ligand-binding domain (LBD). The activator function two domain (AF2) is then stabilized in a conformation favoring recruitment of co-activators, initiating transcriptional regulation. In the context of RXRA/PPAR heterodimers, activation can be induced with agonists to either receptor (*Evans and Mangelsdorf, 2014*). Solved crystal structures of the RXRA/PPARG heterodimer reveal that RXRA S427 resides in helix 10 at a heterodimerization interface (*Figure 3A*) (*Chandra et al., 2008*; *Gampe et al., 2000*). Based on its location, we hypothesized that the RXRA mutation directly activates the PPARG AF2, independent of PPAR ligand binding. To gain evidence, we determined the inducibility of PPARG Q286P, a previously characterized substitution in the ligand-binding pocket (*Figure 3A*) that prevented ligand activation by a panel of PPARG agonists (*Walkey and Spiegelman, 2008*). The Q286P mutation completely blocked pioglitazone-driven receptor activation, but RXRA S427F-driven receptor hyperactivity was maintained (*Figure 3B*). In contrast, the PPARG E471A mutation previously characterized to diminish the transactivation activity of the PPARG AF2 (*Chen et al., 2000*), strongly reduced both pioglitazone and RXRA S427F-driven activity (*Figure 3B*). The analogous AF2 mutation in RXRA, E453A, diminished activation by the RXRA agonist SR11237, but had no impact on mutation-driven hyperactivity (*Figure 3C*). These data support a model in which RXRA$^{S427F}$ hyperactivity relies primarily on the PPAR AF2, independent of PPAR agonist binding.

To gain insight into how RXRA$^{S427F/Y}$ regulates the PPARG AF2, we performed long-time scale molecular simulations of RXRA/PPARG heterodimers with wild-type RXRA or after the S427F substitution. The starting structure was the agonist-bound crystal structure of the RXRA/PPARG LBDs (PDB: 1FM6) (*Gampe et al., 2000*); however, the agonist compounds were deleted to reduce subsequent computational complexity and to enable drift toward the inactivated state. We next

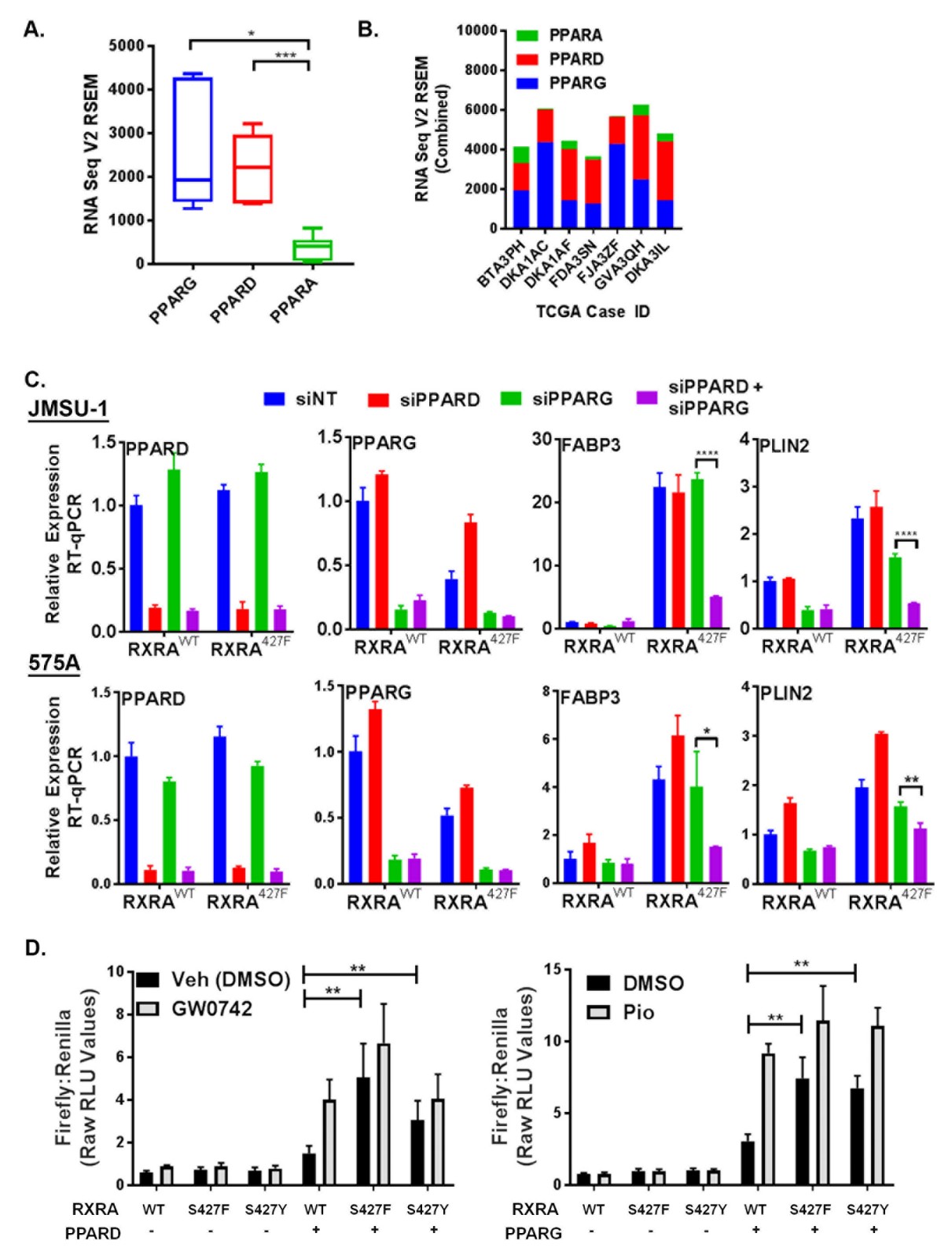

**Figure 2.** PPARG or PPARD expression is necessary and sufficient for mutant RXRA activity. **A)** PPAR RNA expression in RXRA hot-spot mutant clinical samples from the TCGA dataset. Whisker plot shows 25th, median, and 75th percentile. (**B**) Data from panel A plotted per patient with hot-spot mutation. (**C**) Effects of siRNA-mediated knock-down of *PPARD* and *PPARG* in JMSU-1 and 575A cell lines on two target genes (*PLIN2* and *FABP3*) up-regulated by mutant RXRA. Data by RT-qPCR in triplicate ±SD and indicated comparisons by Student's t-test. (**D**) DR1 luciferase reporter activity in UM-

*Figure 2 continued on next page*

*Figure 2 continued*

UC-3 cells transfected with RXRA ±PPARD or PPARG. Cells were treated with 1 μM of the PPARG agonist pioglitazone or the PPARD agonist GW0742 for 16 hr. Data represents mean ±SEM of Firefly to Renilla luciferase signal from three independent experiments done on different days, each performed using triplicate cell wells. Statistical comparisons are by paired *t*-test.

DOI: https://doi.org/10.7554/eLife.30862.006

The following figure supplement is available for figure 2:

**Figure supplement 1.** RARA expression is not sufficient for mutant RXRA hyperactivity.

DOI: https://doi.org/10.7554/eLife.30862.007

constructed Markov State Models (MSMs) following the procedure from *Hart et al. (2016)* to quantify the thermodynamics and kinetics of each heterodimer. First, conformations adopted by backbone-heavy atoms near the mutated site in RXRA and the PPARG AF2 (RXRA 425–429 and PPARG 445–477) from both wild-type and mutant simulations were clustered in the same state space. An MSM was then constructed for each variant based on how often the corresponding simulations transitioned between every pair of clusters. The wild-type and mutant largely populate different clusters, consistent with our hypothesis that mutant and wild-type RXRA differentially regulate the PPARG AF2 region (*Figure 3—figure supplement 1A*)

We next inspected the three most frequently occupied clusters for wild-type and mutant by superimposing the PPARG residues 445–459 on helix 11 as an anchor, enabling us to visualize relative displacement of the AF2 region located in helix 12 (*Figure 3D*). In the RXRA mutant condition, PPARG E471 more closely approximated the agonist confirmation, in agreement with our mutational studies showing the importance of this residue. The terminal tyrosine of PPARG (Y477) also appeared in distinct spaces in the two conditions. Comparing the 5% most frequently occupied clusters for wild type and mutant, we observed a significant difference in inter-residue distance between RXRA 427 and PPARG 477 (*Figure 3E*). These data raised the possibility that an aromatic interaction between RXRA S427F/Y and the terminal tyrosine found in all PPARs may underlie activation of PPAR (*Figure 3F*). To test this, the terminal tyrosine was either deleted or mutated to serine in both PPARG (Y477) and PPARD (Y441) and expression of the mutant PPARs was confirmed by western blot (*Figure 3—figure supplement 1B*). Inducibility by RXRA$^{S427F}$ or agonist was then determined (*Figure 3B and G*). In all cases, inducibility by agonist was maintained, but RXRA S427F-mediated hyperactivity was eliminated. We conclude that S427F/Y initiates an allosteric relay via an aromatic interaction with the terminal tyrosine on PPAR, leading to PPAR AF2 activation. No other RXR dimerization partners have an aromatic amino acid at the corresponding position, providing a structural basis for the selective activation of PPARs (*Figure 3F*).

## PPARD activation induces growth-factor-independent growth of urothelial organoids in the context of tumor suppressor loss

We next sought to determine if PPAR activation drives proliferation of urothelial cells. To culture primary urothelial cells, we developed a mouse urothelial organoid culture system. Similar organoid systems have proven successful at identifying pro-tumorigenic phenotypes driven by cancer-associated mutations in multiple tissue types (*Sachs and Clevers, 2014*; *Karthaus et al., 2014*; *Drost et al., 2015*; *Hwang et al., 2016*; *Boj et al., 2015*). With our urothelial approach, hollow spherical epithelial structures can reliably be grown and passaged from primary mouse bladder epithelium, utilizing epidermal growth factor (EGF) as the primary growth factor (*Figure 4—figure supplement 1A*). Because growth-factor-independent growth is a classical hallmark of transformed cells, we asked if stimulation with PPAR agonists could confer growth in the absence of EGF. As expected, there was no growth of three independently derived wild-type organoids in the absence of growth factor. Treatment with the PPARD agonist GW0742 or the PPARG agonist pioglitazone had no significant effect on growth in these conditions, suggesting that PPAR activation is not sufficient to drive growth in normal urothelial organoids (*Figure 4A*).

Bladder cancer genomes typically harbor mutations in several genes recognized as recurrently altered in bladder cancers (*Cancer Genome Atlas Research Network, 2014*). We speculated that pro-tumorigenic activities of PPAR signaling may only be apparent in the context of tumor suppressor loss. For our analysis, we focused on *TP53* and *KDM6A* which are amongst the most frequently

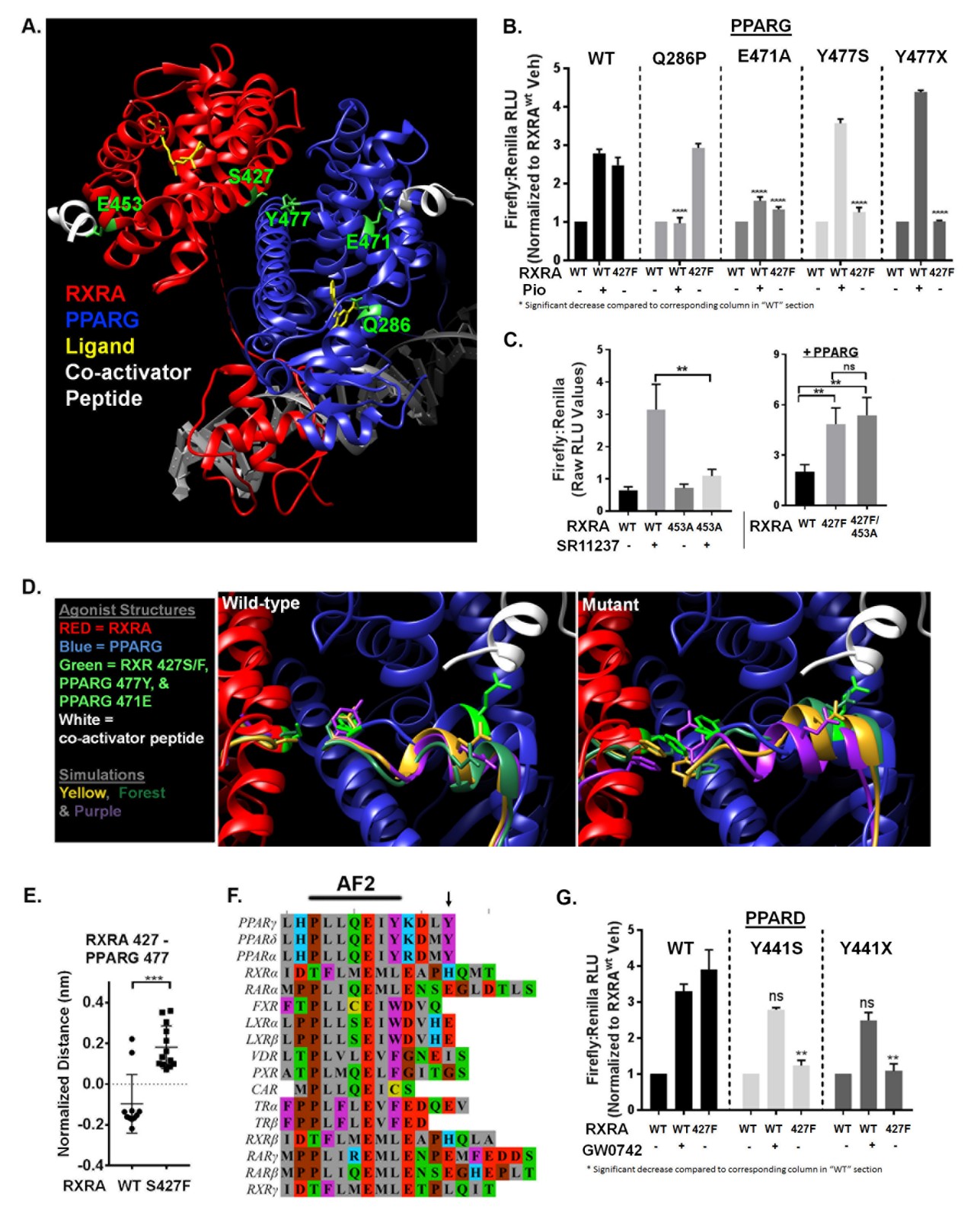

**Figure 3.** Mutant RXRA induces allosteric activation of PPARs through their terminal tyrosine. **A)** RXRA S427 and other amino acids mutated for structure-function studies highlighted in green on a published full-length crystal structure of a RXRA/PPARG heterodimer. (**B**) DR1 reporter assay in UM-UC-3 co-transfecting indicated RXRA and PPARG alleles. Cells were treated with vehicle (DMSO) or pioglitazone 1 µM for 16 hr. Data represents mean normalized signal ±SEM of three independent experiments done on different days, each performed in triplicate, with data from each experiment

*Figure 3 continued on next page*

*Figure 3 continued*
normalized to the RXRA^wt vehicle condition for each section. Statistical comparisons are by unpaired *t*-test. (**C**) Left, reporter assay performed with indicated RXRA alleles only and drug treatment with the RXRA agonist SR11237 (100 nM) for 16 hr. Right, reporter assay with wild-type PPARG co-transfected. Data represent mean ±SEM of Firefly to Renilla luciferase signal from three independent experiments done on different days, each performed using triplicate cell wells. Statistical comparisons are by paired *t*-test. (**D**) Published agonist structure of RXRA/PPARG heterodimer (PDB: 1FM6) in red and blue with key residues highlighted in bright green. Top three occupied microstate clusters from simulation experiments are superimposed. (**E**) Distance from starting agonist structure between alpha carbons of RXR 427 and PPARG 477 in the top 5% most-occupied microstates for wild-type and mutant RXRA. Mean ±SD, comparison is by Student's t test. (**F**) Alignment of the AF2 region and C-terminus of all RXRA dimerization partners. Terminal tyrosine unique to PPARs is indicated. (**G**) Reporter assay similar to B, but using PPARD and the PPARD agonist GW0742.
DOI: https://doi.org/10.7554/eLife.30862.008
The following figure supplement is available for figure 3:

**Figure supplement 1.**
DOI: https://doi.org/10.7554/eLife.30862.009

mutated tumor suppressors in bladder cancer, mutated in ~50% and~25% of bladder cancer cases, respectively (*Cancer Genome Atlas Research Network, 2014*; *Guo et al., 2013*; *Van Allen et al., 2014*). Of 16 patient samples identified with an *RXRA* hot-spot mutation across three cohorts, four had mutations in both *TP53* and *KDM6A*, five had mutations in *TP53* only, three had mutations in *KDM6A* only, and four had mutations in neither (*Cerami et al., 2012*). To determine if loss of tumor suppressor function would enable pro-tumorigenic activities of PPAR activation, we generated conditional knock-out bladder organoids from mice with the following three genotypes: (1) *Trp53^{F/F}*, (2) *Kdm6a^F*, and (3) *Trp53^{F/F};Kdm6a^F*. Three independent organoid lines (each from a distinct mouse bladder) were generated with each genotype. These organoid lines were then all infected with Adeno-cre and complete deletion of the conditional alleles was confirmed by genotyping PCR (not shown). The wild-type organoids utilized in *Figure 4A* were generated from the littermates of these mice and also were infected with Adeno-Cre to control for Cre exposure.

We then determined growth of the organoids in EGF deplete media after treatment with PPAR agonists (*Figure 4B*, *Figure 4—figure supplement 1B*). No genotype showed consistent positive growth in the vehicle condition. Elimination of *Trp53*, but not *Kdm6a*, was sufficient for GW0742 to drive proliferation. With combined knock-out of *Trp53* and *Kdm6a*, growth induction by GW0742 was even more robust than with *Trp53* knock-out alone. These data suggest that loss of tumor suppressors creates a context permissive for PPARD-driven proliferation. Even though responsiveness to GW0742 was consistent across all of the dual knock-out organoids, we reasoned that utilizing a clonal sub-line might minimize variation due to random clonal skewing in later experiments involving retroviral infection and drug selection. Therefore, we established DKO 431.A from a single organoid picked from the DKO 431 line and assessed its responsiveness to PPAR agonists. DKO 431.A showed strong, dose-dependent, enhanced growth when treated with GW0742, but not pioglitazone (*Figure 4C*). To assess transcriptional activation by the same PPAR agonists, DKO 431.A organoids were plated in standard media for 7 days and then treated with two doses of GW0742 or pioglitazone for 48 hr. Expression of two PPAR target genes, *Plin2* and *Fabp4* was then determined by RT-qPCR. GW0742 strongly induced both genes, while pioglitazone and little to no effect (*Figure 4D*, *Figure 4—figure supplement 1C*). These findings confirm that the ability of PPAR agonists to induce target genes in DKO 431.A correlates with their ability to promote growth-factor-independent growth. The apparent lack of response to PPARG agonist in these experiments is likely simply due to its lower expression in these mouse organoids relative to 575A and JMSU-1, the two cells lines in which we had established induction of PPAR target genes with pioglitazone in *Figure 1* (*Figure 4—figure supplement 1D*).

## Mutant RXRA drives sustained growth-factor-independent growth in urothelial organoids with concurrent loss of Kdm6a and Trp53

We then asked if RXRA^{S427F} would promote growth-factor-independent growth. DKO 431.A was infected with retrovirus bearing empty vector, RXRA^wt, or RXRA^{S427F} and similar expression levels of RXRA between the wild-type and mutant condition was confirmed by western blot and qPCR (*Figure 4E*). Growth of the organoids in EGF deplete media was then assessed. Mutant RXRA-

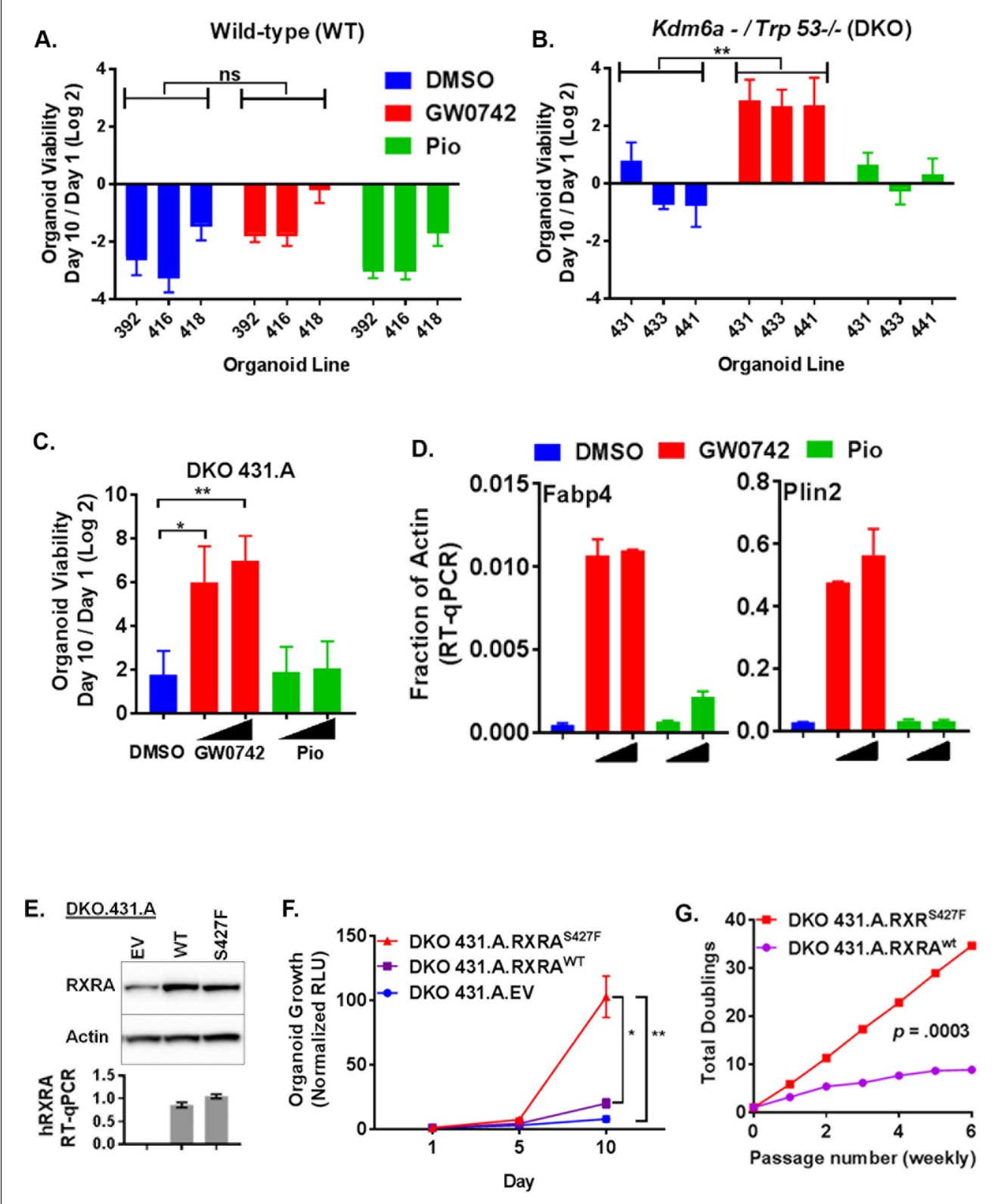

**Figure 4.** PPARD agonist and mutant RXRA confer growth-factor-independent growth to urothelium in the context of tumor suppressor loss. (**A**) Organoids were derived from three independent wild-type mouse bladders and infected with Adeno-Cre. Growth was determined for each line in organoid media without EGF using CellTiter-Glo after treatment with vehicle, GW0742 (100 nM), or pioglitazone (100 nM). Data represent mean ±SEM from three independent experiments, each performed in triplicate cell wells. Statistical comparison is by unpaired *t*-test. (**B**) Similar to A but organoids
*Figure 4 continued on next page*

*Figure 4 continued*

were derived from Trp53$^{flox/flox}$; Kdm6a$^{flox}$ mice. (C) CellTiter-Glo growth assay in media without EGF of a sub-clone from DKO431 (DKO 431.A) treated with GW0742 (10, 100 nM) or pioglitazone (100, 1000 nM). Mean values ± SEM from three independent experiments, each performed using triplicate organoid wells. Comparison is by paired *t*-test. (D) DKO 431.A organoids were plated in standard organoid media and then treated with, vehicle, GW0742 (10, 100 nM)) or pioglitazone (100, 1000 nM) for an additional 48 hr. Induction of PPAR target genes was determined by RT-qPCR in triplicate ± SD. (See also *Figure 4—figure supplement 1C*.) (E) DKO 431.A organoids were infected with a retroviral vector that was empty, expresses RXRA$^{wt}$, or expresses RXRA$^{S427F}$ and expression of total RXRA (western blot) and human *RXRA* (RT-qPCR, triplicate +/SD) was determined. (F) Mean CellTiter-Glo signal ±SEM from three identical experiments, each performed using triplicate organoid wells. Comparison of D10 data is by paired *t*-test. (G) Identical number of indicated organoid cells were plated in media without EGF and then harvested with trypsin weekly and counted in duplicate using a BioRad TC20. Identical numbers of cells were then re-plated and this was repeated for 6 weeks. Total cell number doublings were calculated and plotted. Comparison of doublings is by paired *t*-test.

DOI: https://doi.org/10.7554/eLife.30862.010

The following figure supplement is available for figure 4:

**Figure supplement 1.**

DOI: https://doi.org/10.7554/eLife.30862.011

expressing organoids grew significantly faster than empty vector or wild-type expressing organoids (*Figure 4F*). However, growth was also observed in the wild-type condition. To ensure that the difference between mutant and wild-type growth was robust, we passaged wild-type and mutant-expressing organoids with tyrpsinization weekly, always plating an identical number of cells and then counting the number of cells present at time of each passage. The total number of doublings over six passages was then determined. Mutant RXRA organoids showed consistent growth throughout the experiment, while the wild-type condition appeared to plateau (*Figure 4G*). Comparing the number of doublings per passage between the two conditions, we found the difference to be highly significant (paired t test p=0.0003), confirming gain-of-function with the mutant allele.

## Mutant RXRA-driven proliferation of urothelial organoids is dependent on PPARD activity

Having established that RXRA$^{S427F}$ phenocopies GW0742 induced growth in the organoid growth assay, we next looked for transcriptional evidence that RXRA$^{S427F}$ was activating PPARD. Organoids were cultured in standard media for seven days and then treated with PPAR antagonists for an additional two days. We then determined mRNA expression levels of two PPAR target genes by RT-qPCR. Comparing vehicle-treated samples, *Plin2* and *Fabp4* were found significantly up-regulated in the mutant condition compared to the empty vector or wild-type organoids, confirming that mutant RXRA up-regulates expected PPAR targets in the organoid system (*Figure 5A*). Treatment of the mutant expressing organoids with either of two PPARD inverse agonists, ST247 and GSK0660, blunted up-regulation of the target genes (*Figure 5A*, *Figure 5—figure supplement 1*) (*Naruhn et al., 2011*; *Shearer et al., 2008*). The PPARG antagonist T0070907 did not show any inhibitory activity. These findings confirm that mutant RXRA drives PPARD hyper-activity in the organoid system.

We next sought to determine if mutant RXRA-driven growth could be reversed with PPAR inhibition. When treated with either of the two PPARD antagonists, the mutant expressing organoids showed significant, dose-dependent growth inhibition (*Figure 5B*.) To determine if the proliferative effects of PPARD inhibitors was specific to the RXRA-mutant-driven proliferation, we utilized an organoid line we established from a carcinogen-induced bladder tumor, called MCB6C, which shows robust growth in EGF deplete media, but does not harbor the RXRA mutation (not shown). Treatment of MCB6C with the PPARD inhibitors in EGF deplete media did not inhibit their growth across the identical doses (*Figure 5C*). As expected based on the gene expression analysis, the PPARG antagonist T0070907 also had no antiproliferative effects on 431.A.RXRA$^{S427F}$ (*Figure 5D*). Lastly, 431.A.RXR$^{wt}$ organoids in EGF-containing media were treated with the same PPAR antagonists, none of which significantly inhibited growth of this organoid line (*Figure 5E*), confirming PPAR dependence is specific to mutant RXRA-driven proliferation.

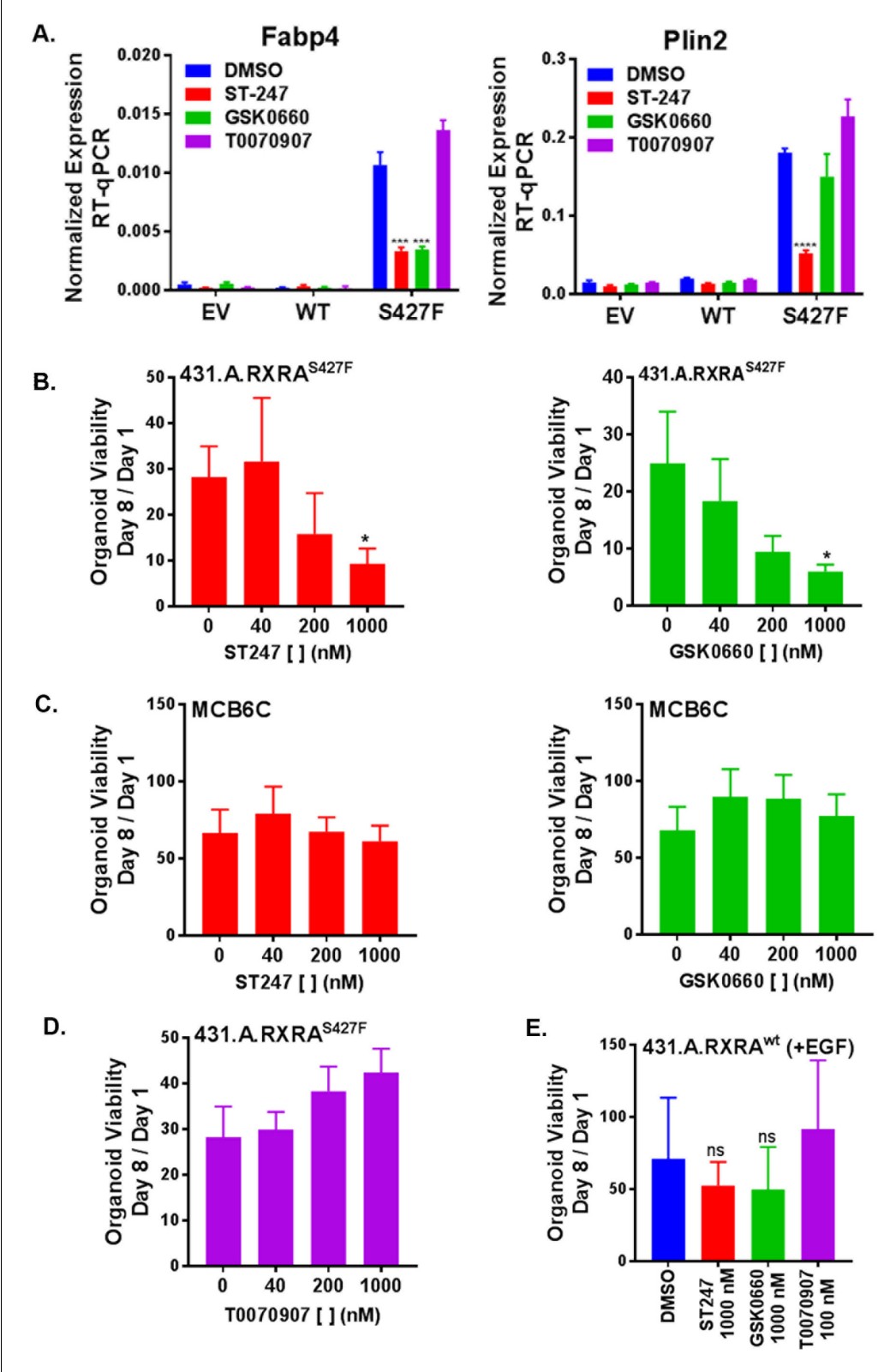

**Figure 5.** RXRA S427F generates PPARD-dependent urothelial growth. (**A**) Retrovirally transduced organoids from *Figure 4* were plated for 7 days in standard media and then treated with the indicated PPARD antagonists (1000 nM ST-247, GSK0660) or PPARG antagonist (100 nM T0070907) for 2 days. Expression of PPAR targets was determined by RT-qPCR in triplicate ±SD and comparison is by Student's t-test to the RXRA^S427F DMSO condition. (See also *Figure 5—figure supplement 1*)) (**B-E**) CellTiter-Glo growth assay of indicated organoid lines treated with indicated drugs. Plotted is mean

*Figure 5 continued on next page*

*Figure 5 continued*

signal ±SEM from three independent experiments, each performed using triplicate organoid wells. MCB6C is an organoid line we derived from a carcinogen-induced bladder tumor and lacks RXRA mutation. Organoids were cultured in media without EGF except for panel E where inclusion of EGF is indicated. Comparison is to DMSO condition using paired *t*-test.

DOI: https://doi.org/10.7554/eLife.30862.012

The following figure supplement is available for figure 5:

**Figure supplement 1.** Aggregated RT-qPCR from three distinct experiments as described in *Figure 5A* with each data point from each experiment plotted with mean indicated.

DOI: https://doi.org/10.7554/eLife.30862.013

## Discussion

A role for PPAR activity in promoting bladder cancer is supported by the presence of a *PPARG* gene amplification in 17% of cases (*Cancer Genome Atlas Research Network, 2014*). Intriguingly, pioglitazone use in patients with diabetes appears to increase bladder cancer risk, further hinting that PPARG activation can promote bladder cancer growth (*Lewis et al., 2011*). Our studies identified RXRA point mutations as a second genomic mechanism by which PPARs can be hyperactivated. These data, along with our direct functional evidence that PPAR activation by agonist or mutant RXRA drives urothelial proliferation, build on a growing body of data credentialing PPARs as bladder cancer drivers. Importantly, human bladder cancer samples appear to express both *PPARD* and *PPARG* and our data suggest both are activated by mutant RXRA and play a role in transcriptional hyperactivity in human bladder cancer cells. In order to guide drug development, future studies will be needed with a broad panel of human-derived tissue to determine if one PPAR isoform is of greater clinical relevance to RXRA mutation-driven disease. We speculate co-targeting of PPARD and PPARG will be necessary, perhaps even in *PPARG* amplified cases, as functional redundancy of homologous nuclear receptors or kinases is known to be exploited by tumors, in some cases underlying treatment resistance to therapeutics targeting only one homologue (*Arora et al., 2013*; *Juric et al., 2015*).

Heyman and colleagues showed two decades ago that RXR agonists can induce activation of RXR/PPAR heterodimers, independent of the RXR AF2, by inducing conformational changes in PPAR (*Schulman et al., 1998*). The RXRA hot-spot mutations appear to co-opt a similar, although mechanistically distinct, activation principle, relying on the introduction of an aromatic interaction that drives ligand-independent activation of PPARs. While this manuscript was under preparation, Korpal et al. reported biophysical data characterizing the mutant RXRA/PPARG heterodimer (*Korpal et al., 2017*). Those in vitro findings align with our modeling predictions, which we further functionally validated with cell-based mutational studies, both with PPARG and PPARD. This allosteric activation mechanism of the PPAR AF2 by mutant RXRA raises unique challenges with regard to inhibitor design. In our study, we utilized inverse agonists which bind the LBD and stabilize interactions with co-repressors. While they are effective at reversing mutant RXRA-driven activity, we suspect chemical screens using assays in which PPARs are activated through the mutation-based mechanism may identify optimal inverse agonists capable of more complete target inhibition.

Inverse agonists binding the PPAR LBD remain the most characterized strategy to inhibit PPAR activity, but may not prove to be the most effective approach for blocking ligand independent activity driven by RXRA mutations. A potentially more effective strategy might be through the identification of small molecules that induce PPAR degradation. Supporting the concept, ligands that induce destabilizing conformations in ER have been identified (*Wu et al., 2005*), and the clinical utility of the ER degrader fluvestrant has been validated in breast cancer clinical trials (*Howell et al., 2002*; *Osborne et al., 2002*). Similar to ER, PPARs undergo agonist-induced degradation, suggesting therapeutic ligand-induced degradation of PPARs might also be feasible (*Hauser et al., 2000*). Perhaps, a more rational approach to inducing PPAR degradation would be to engineer PPAR ligands linked to a chemical tag (e.g. PROTAC) that targets proteins to the proteasome. Such approaches have been used in preclinical systems to degrade other nuclear receptors (*Rodriguez-Gonzalez et al., 2008*), although the clinical utility of the approach is not yet established. Lastly, the identification of biologically active small molecules that disrupt co-regulator binding to the ER AF2 raises yet another possible paradigm for targeting ligand-independent activation through the identification of small

molecules that recognize the activated PPAR AF2 (*Raj et al., 2017*). Our work here should encourage such approaches with the goal of identifying inhibitors that effectively overcome PPAR tumorigenic activity due to *PPARG* amplification and *RXRA* mutation.

The organoid system provided us with a unique capability to assay epithelial autonomous effects of PPAR activity in defined genetic contexts, including normal urothelium. In contrast to classical growth factors such as EGF which stimulate proliferation of wild-type organoids, we found that PPARD activation is only sufficient to drive growth with concurrent tumor suppressor loss. One possible explanation is that tumor suppressor loss alters PPARD-driven transcriptional regulation. Studies with AR in prostate cancer suggest that the AR cistrome is broadly reprogrammed during tumorigenesis (*Pomerantz et al., 2015*), supporting a model in which nuclear receptors take on de novo target regulation in transformed cells. Alternatively, tumor suppressor loss may alter the cellular response to PPARD activation and/or provide complementary pro-tumorigenic signals. These possibilities will be explored in future studies and should help elucidate the cellular mechanisms by which PPAR signaling can drive urothelial proliferation. Regardless of the underlying mechanisms, however, our findings, along with recent work from others demonstrating PPARG orchestrates an immunosuppressive micro-environment to enhance tumor growth in vivo (*Korpal et al., 2017*), provide an increasingly compelling rationale for the development of a new class of molecularly targeted drugs with potential relevance to at least a quarter of bladder cancer cases.

# Materials and methods

## Key resources table

| Reagent type (species) or resource | Designation | Source or reference | Identifiers | Additional information |
|---|---|---|---|---|
| gene (Homo sapiens) | RXRA | NA | NCBI Gene ID:6256; NM_002957 | |
| gene (H. sapiens) | PPARG | NA | NCBI Gene ID:5468; NM_138711 | |
| gene (H. sapiens) | PPARD | NA | NCBI Gene ID:5467; NM_006238 | |
| strain, strain background (Mus musculus) | Kdm6a$^F$ | other | | Generated by Dr. Lukas Wartman (Washington University School of Medicine) with ES cells obtained from EUCOMM with the Kdm6a$^{tm1a (EUCOMM)Wtsi}$ allele (manuscript in preparation) |
| strain, strain background (M. musculus) | Trp53Flox; B6.129P2-Trp53tm1brn/J | The Jackson Laboratory | The Jackson Laboratory:008462; RRID:IMSR_JAX:008462 | |
| genetic reagent | Ad5CMVCre-eGFP adenovirus | University of Iowa Viral Vector Core | VVC-U of Iowa:1174 | |
| cell line (M. musculus) | MCB6C | this paper | | Clonal organoid line generated from tumor bearing bladder of male C57BL/6 mouse treated with BBN |
| cell line (M. musculus) | DKO 431.A | this paper | | clonal organoid line generated from the urothelium of a male Trp53$^{Flox/Flox}$;Kdm6a$^{Flox}$ mouse |
| cell line (M. musculus) | WT | this paper | | Organoid lines generated from the urothelium of wild-type male mice resulting from cross between Trp53$^{Flox/+}$ and Kdm6a$^{Flox/+}$ mice |
| cell line (M. musculus) | "Trp53-/-; Kdm6a-"; DKO | this paper | | Organoid lines were generated from the urothelium of Trp53$^{Flox/Flox}$; Kdm6a$^{Flox}$ male mice and then infected with Ad5CMVCre-eGFP adenovirus in vitro |

*Continued on next page*

*Continued*

| Reagent type (species) or resource | Designation | Source or reference | Identifiers | Additional information |
|---|---|---|---|---|
| cell line (M. musculus) | "Kdm6a-"; KKO | this paper | | Organoid lines were generated from the urothelium of Kdm6a$^{Flox}$ male mice and then infected with Ad5CMVCre-eGFP adenovirus in vitro |
| cell line (M. musculus) | "Trp53-/-"; PKO | this paper | | Organoid lines were generated from the urothelium of Trp53$^{Flox/Flox}$ male mice and then infected with Ad5CMVCre-eGFP adenovirus in vitro |
| cell line (M. musculus) | DKO 431.A.EV | this paper | | DKO 431.A organoid line infected with retrovirus carrying pBABE puro empty vector |
| cell line (M. musculus) | DKO 431.A.RXRAwt | this paper | | DKO 431.A organoid line infected with retrovirus carrying pBABE puro RXRA |
| cell line (M. musculus) | DKO 431.A.RXRAS427F | this paper | | DKO 431.A organoid line infected with retrovirus carrying pBABE puro RXRA S427F |
| cell line (H. sapiens) | JMSU-1 | other | RRID:CVCL_2081 | obtained from Dr. David Solit (MSKCC) |
| cell line (H. sapiens) | 575A | other | RRID:CVCL_7941 | obtained from Dr. David Solit (MSKCC) |
| cell line (H. sapiens) | UM-UC-3 | other | RRID:CVCL_1783 | obtained from Dr. David Solit (MSKCC) |
| cell line (H. sapiens) | Lenti-X 293T | Clontech | Clontech:632180 | |
| cell line (H. sapiens) | JMSU-1 RXRA WT | this paper | | JMSU-1 cell line infected with retrovirus carrying pBABE puro RXRA |
| cell line (H. sapiens) | JMSU-1 RXRA S427F | this paper | | JMSU-1 cell line infected with retrovirus carrying pBABE puro RXRA S427F |
| cell line (H. sapiens) | JMSU-1 RXRA S427Y | this paper | | JMSU-1 cell line infected with retrovirus carrying pBABE puro RXRA S427Y |
| cell line (H. sapiens) | 575A RXRA WT | this paper | | 575A cell line infected with retrovirus carrying pBABE puro RXRA |
| cell line (H. sapiens) | 575A RXRA S427F | this paper | | 575A cell line infected with retrovirus carrying pBABE puro RXRA S427F |
| antibody | anti-PPARG (81B8) (rabbit monoclonal) | Cell Signaling Technology | Cell Signaling Technology:2443; RRID:AB_823598 | (1:1000) |
| antibody | anti-PPARD (rabbit monoclonal) | Abcam | Abcam:ab178866 | (1:5000) |
| antibody | anti-RXRA (D6H10) (rabbit monoclonal) | Cell Signaling Technology | Cell Signaling Technology:3085 | (1:1200) |
| antibody | anti-beta-Actin (mouse monoclonal) | Sigma-Aldrich | Sigma-Aldrich:A5441; RRID:AB_476744 | (1:50000) |
| antibody | anti-GAPDH (D16H11) (rabbit monoclonal) | Cell Signaling Technology | Cell Signaling Technology:5174; RRID:AB_10622025 | (1:1000) |
| antibody | anti-rabbit IgG, HRP (goat) | Cell Signaling Technology | Cell Signaling Technology:7074; RRID:AB_2099233 | (1:7500) |

*Continued on next page*

*Continued*

| Reagent type (species) or resource | Designation | Source or reference | Identifiers | Additional information |
|---|---|---|---|---|
| antibody | anti-mouse IgG, HRP (horse) | Cell Signaling Technology | Cell Signaling Technology:7076; RRID:AB_330924 | (1:7500) |
| antibody | anti-RXRA (K8508) (mouse monoclonal) | R&D Systems | R&D Systems: PP-K8508-00; RRID:AB_2182738 | (5 µg) |
| antibody | anti-H3K27Ac (rabbit polyclonal) | Abcam | Abcam:ab4729; RRID:AB_2118291 | (0.4 µg) |
| recombinant DNA reagent | PPRE X3-TK-luc; DR1 reporter (plasmid) | Addgene; PMID 9539737 | Addgene:1015 | plasmid was deposited by Bruce Spiegelman |
| recombinant DNA reagent | pGL3-RARE-luciferase; DR5 reporter (plasmid) | Addgene; PMID 16818722 | Addgene:13458 | plasmid was deposited by T. Michael Underhill |
| recombinant DNA reagent | pRL-SV40 (plasmid) | Promega | Promega:E2231 | |
| recombinant DNA reagent | pCL-ampho (plasmid) | other | | obtained from Dr. Charles Sawyers (MSKCC) |
| recombinant DNA reagent | VSVG (plasmid) | other | | obtained from Dr. Charles Sawyers (MSKCC) |
| recombinant DNA reagent | pCMV6-XL4 RARA (plasmid) | OriGene | OriGene:SC119566 | |
| recombinant DNA reagent | pCMV6-XL4 PPARG (plasmid) | OriGene | OriGene:SC108192 | |
| recombinant DNA reagent | pCMV6-XL4 PPARG Q286P (plasmid) | this paper | | Q286 was mutated via site-directed mutagenesis of pCMV6-XL4 PPARG |
| recombinant DNA reagent | pCMV6-XL4 PPARG E471A (plasmid) | this paper | | E471 was mutated via site-directed mutagenesis of pCMV6-XL4 PPARG |
| recombinant DNA reagent | pCMV6-XL4 PPARG Y477S (plasmid) | this paper | | Y477 was mutated via site-directed mutagenesis of pCMV6-XL4 PPARG |
| recombinant DNA reagent | pCMV6-XL4 PPARG Y477X (plasmid) | this paper | | Y477 was deleted via site-directed mutagenesis of pCMV6-XL4 PPARG |
| recombinant DNA reagent | pCMV6-XL4 empty vector (plasmid) | this paper | | generated by digesting pCMV6-XL4 PPARG with NotI to remove PPARG and by ligating the plasmid ends with T4 DNA ligase |
| recombinant DNA reagent | pBABE puro RXRA (plasmid) | Addgene | Addgene:11441 | deposited by Ronald Kahn |
| recombinant DNA reagent | pBABE puro RXRA S427F (plasmid) | this paper | | S427 was mutated via site-directed mutagenesis of pBABE puro empty vector |
| recombinant DNA reagent | pBABE puro RXRA S427Y (plasmid) | this paper | | S427 was mutated via site-directed mutagenesis of pBABE puro empty vector |
| recombinant DNA reagent | pBABE puro RXRA E453A (plasmid) | this paper | | E453 was mutated via site-directed mutagenesis of pBABE puro empty vector |
| recombinant DNA reagent | pBABE puro RXRA S427F/E453A (plasmid) | this paper | | E453 was mutated via site-directed mutagenesis of pBABE puro RXRA S427F |
| recombinant DNA reagent | pBABE puro empty vector (plasmid) | this paper | | generated by digesting pBABE puro RXRA with EcoRI to remove RXRA and by ligating the plasmid ends with T4 DNA ligase |
| recombinant DNA reagent | pCMV6-XL4 PPARD (plasmid) | this paper | | Human PPARD was cloned from JMSU-1 epithelial bladder cancer cells and inserted into the pCMV6-XL4 |
| recombinant DNA reagent | pCMV6-XL4 PPARD Y441S (plasmid) | this paper | | Y441 was mutated via site-directed mutagenesis of pCMV6-XL4 PPARD |

*Continued on next page*

*Continued*

| Reagent type (species) or resource | Designation | Source or reference | Identifiers | Additional information |
|---|---|---|---|---|
| recombinant DNA reagent | pCMV6-XL4 PPARD Y441X (plasmid) | this paper | | Y441 was deleted via site-directed mutagenesis of pCMV6-XL4 PPARD |
| sequence-based reagent | ON-TARGETplus Non-targeting Pool (siRNA) | Dharmacon | Dharmacon: D-001810-10-20 | |
| sequence-based reagent | ON-TARGETplus Human PPARG siRNA | Dharmacon | Dharmacon: L-003436-00-0005 | |
| sequence-based reagent | ON-TARGETplus Human PPARD siRNA | Dharmacon | Dharmacon: L-003435-00-0005 | |
| commercial assay or kit | Dual-Glo Luciferase Assay System | Promega | Promega:E2940 | |
| commercial assay or kit | CellTiter-Glo | Promega | Promega:G7571 | |
| commercial assay or kit | Ovation Ultraflow System V2 | NuGen | NuGen:0344-32 | |
| chemical compound, drug | Pioglitazone | Sigma-Aldrich | Sigma-Aldrich:E6910 | |
| chemical compound, drug | GW 0742 | Tocris | Tocris:2229 | |
| chemical compound, drug | SR 11237 | Tocris | Tocris:3411 | |
| chemical compound, drug | all-trans-Retinoic Acid (ATRA) | Sigma-Aldrich | Sigma-Aldrich:R2625 | |
| chemical compound, drug | GSK 0660 | Tocris | Tocris:3433 | |
| chemical compound, drug | ST247 | Sigma-Aldrich | Sigma-Aldrich:SML0424 | |
| chemical compound, drug | T0070907 | Cayman Chemical | Cayman Chemical:10026 | |
| software, algorithm | GROMACS 5.1.3 | DOI: 10.1016/j.softx. 2015.06.001 | RRID:SCR_014565 | |
| software, algorithm | MSMBuilder 2.8 | PMID: 22125474 | | |
| software, algorithm | Chimera | PMID: 15264254; http://www.rbvi.ucsf. edu/chimera | RRID:SCR_004097 | |
| sequence-based reagent | mKDM6A Forward (primer) | this paper | | 5' CGAGAAAGGAAATGTG AGAGCAAGG 3' |
| sequence-based reagent | mKDM6A Reverse 4 (primer) | this paper | | 5' CTGGCAGGATATGATA GCAATGTG 3' |
| sequence-based reagent | oIMR8543 (primer) | The Jackson Laboratory; https://www2.jax.org/ protocolsdb/f?p=116: 2:0::NO:2:P2_MASTER_ PROTOCOL_ID,P2_JRS_ CODE:3226,008462 | | 5' GGTTAAACCCAGCT TGACCA 3' |
| sequence-based reagent | oIMR8544 (primer) | The Jackson Laboratory; https://www2.jax.org/ protocolsdb/f?p=116: 2:0::NO:2:P2_MASTER_ PROTOCOL_ID,P2_JRS_ CODE:3226,008462 | | 5' GGAGGCAGAGACA GTTGGAG 3' |
| sequence-based reagent | PPARD.qPCR.Fwd.1 (primer) | this paper | | 5' ATGCACCAACGA GGCTGATG 3' |
| sequence-based reagent | PPARD.qPCR.Rev.1 (primer) | this paper | | 5' CTGCTCCATGGCT GATCTCC 3' |
| sequence-based reagent | PPARG fwd1 (primer) | this paper | | 5' ATGCCTTGCAGT GGGGATGTC 3' |
| sequence-based reagent | PPARG rev1 (primer) | this paper | | 5' GAGGTCAGCGGA CTCTGGATTC 3' |

*Continued on next page*

*Continued*

| Reagent type (species) or resource | Designation | Source or reference | Identifiers | Additional information |
|---|---|---|---|---|
| sequence-based reagent | hPLIN2 fwd1 (primer) | this paper | | 5' AGTGCTCTGCCC ATCATCCAG 3' |
| sequence-based reagent | hPLIN2 rev1 (primer) | this paper | | 5' TCACAGCGCCTT TGGCAT TG 3' |
| sequence-based reagent | FABP4 fwd1 (primer) | this paper | | 5' ACTGCAGCTTCCT TCTCACCTTG 3' |
| sequence-based reagent | FABP4 rev1 (primer) | this paper | | 5' TGCCAGCCACTT TCCTGGTG 3' |
| sequence-based reagent | mPlin2 Fwd1 (primer) | this paper | | 5' GTGCCCTGCCC ATCATCC 3' |
| sequence-based reagent | mPlin2 Rev1 (primer) | this paper | | 5' TTACGGCACCTCT GGCACTG 3' |
| sequence-based reagent | mFabp4 Fwd1 (primer) | this paper | | 5' TGCAGCCTTTCTCA CCTGGAAG 3' |
| sequence-based reagent | mFabp4 Rev1 (primer) | this paper | | 5' GCCTGCCACTTTCC TTGTGG 3' |
| sequence-based reagent | RXRA fwd1 (primer) | this paper | | 5' ACAAGACGGAGC TGGGCTG 3' |
| sequence-based reagent | RXRA rev2 (primer) | this paper | | 5' GGCTGCTCTGGGT ACTTGTGC 3' |
| sequence-based reagent | RXRA E453A SDM For (primer) | this paper | | 5' acaccttccttatggccat gctggaggcgccg 3' |
| sequence-based reagent | RXRA E453A SDM Rev (primer) | this paper | | 5' cggcgcctccagcatggcc ataaggaaggtgt 3' |
| sequence-based reagent | RXRa S427F-F (primer) | this paper | | 5' CCG GCT CTG CGC TTT ATC GGG CTC AAA T 3' |
| sequence-based reagent | RXRa S427F-R (primer) | this paper | | 5' CAT TTG AGC CGG ATA AAG CGC AGA GCC G 3' |
| sequence-based reagent | RXRa S427Y-F (primer) | this paper | | 5' CCG GCT CTG CGC TAT ATC GGG CTC AAA T 3' |
| sequence-based reagent | RXRa S427Y-R (primer) | this paper | | 5' CAT TTG AGC CGG ATA TAG CGC AGA GCC G 3' |
| sequence-based reagent | F hPPARGQ286P (primer) | this paper | | 5' ccacggagcgaaacgg gcagccctgaaag 3' |
| sequence-based reagent | R hPPARGQ286P (primer) | this paper | | 5' ctttcagggctgcccgttt cgctccgtgg 3' |
| sequence-based reagent | F hPPARGE471A (primer) | this paper | | 5' agtccttgtagatcgcctg caggagcggg 3' |
| sequence-based reagent | R hPPARGE471A (primer) | this paper | | 5' cccgctcctgcaggcgat ctacaaggact 3' |
| sequence-based reagent | PPARG Y477S For SDM (primer) | this paper | | 5' GAGATCTACAAGGACTTGAG CTAGCAGAGAGTCCTGAGC 3' |
| sequence-based reagent | PPARG Y477S Rev SDM (primer) | this paper | | 5' GCTCAGGACTCTCTGCTAGCT CAAGTCCTTGTAGATCTC 3' |
| sequence-based reagent | PPARG Y477X For SDM (primer) | this paper | | 5' GATCTACAAGGACTTGTAG TAGCAGAGAGTCCTGA 3' |
| sequence-based reagent | PPARG Y477X Rev SDM (primer) | this paper | | 5' TCAGGACTCTCTGCTACTAC AAGTCCTTGTAGATC 3' |
| sequence-based reagent | PPARD Y441S For (primer) | this paper | | 5' AGATCTACAAGGACATGAG CTAACGGCGGCACCCAG 3' |
| sequence-based reagent | PPARD Y441S Rev (primer) | this paper | | 5' CTGGGTGCCGCCGTTA GCTCATGTCCTTGTAGATCT 3' |

*Continued on next page*

*Continued*

| Reagent type (species) or resource | Designation | Source or reference | Identifiers | Additional information |
|---|---|---|---|---|
| sequence-based reagent | PPARD Y441X For (primer) | this paper | | 5' GATCTACAAGGACATGTGA TAACGGCGGCACCCAGG 3' |
| sequence-based reagent | PPARD Y441X Rev (primer) | this paper | | 5' CCTGGGTGCCGCCGTTATC ACATGTCCTTGTAGATC 3' |

*Mouse Bladder Organoid Generation and Culturing:* Urothelium was dissected from bladders collected from 6- to 8-week-old male mice, minced into smaller pieces with scissors, and digested with collagenase type II (17101015 Gibco) re-suspended at 5 mg/mL in Advanced DMEM/F12+++ medium (advanced DMEM/F-12 medium (12634028 Gibco) supplemented with 1% penicillin-streptomycin and 1% HEPES (MT25060CI Corning)), 5 µM ROCK1/2 inhibitor Y-27632 (72302 Stemcell Technologies), and 0.2 mg/mL elastase (E7885 Sigma) for 4 hr at 37°C while shaking at 700 rpm. After digestion with collagenase, cells were pelleted at 500 x g for 5 min and digested with TrypLE (12605010 Gibco) for 30 min at 37°C with shaking at 700 rpm. Cells were washed with Advanced DMEM/F12+++ medium and re-suspended in growth factor reduced Matrigel (356231 Corning) at 10,000 cells per a 50 µL Matrigel tab. Tabs were incubated at 37°C for 15 min to allow Matrigel to harden. Once hardened, tabs were cultured in organoid medium prepared as described in *Gao et al. (2014)* except for the following changes: final concentration of EGF was 5 ng/µL, final concentration of A83-01 was 20 nM, and FGF10, FGF2, dihydrotestosterone, Y-27632, SB202190, and primocin were omitted. Organoids were split approximately every 7 days. Tabs were homogenized by pipetting. Organoids were digested with TrypLE for approximately 20 min at 37°C with periodic vortexing. Cells were washed with Advanced DMEM/F12+++ medium and re-suspended at 10,000 cells per a 50 µL Matrigel tab. To generate MCB6C, mice were treated with BBN 0.1% via drinking water for 22 weeks. Bladders were harvested and a polyclonal organoid line (MCB6) was generated from a tumor-bearing bladder. A single organoid was isolated to generate MCB6C.

## Mice

Animals were handled and housed according to protocols approved by the Washington University School of Medicine Institutional Animal Care and Use Committee. Kdm6a[F] mice were generated by Dr. Lukas Wartman (Washington University School of Medicine) with ES cells obtained from EUCOMM with the Kdm6a[tm1a(EUCOMM)Wtsi] allele (manuscript in preparation) and maintained on the C57BL6/J background. B6.129P2-Trp53[tm1brn]/J (Trp53[Flox]) mice were purchased from Jackson Laboratory (stock # 008462). Mice were genotyped via PCR (Kdm6a 5' CGAGAAAGGAAATGTGAGAG-CAAGG 3' and 5' CTGGCAGGATATGATAGCAATGTG 3'; Trp53 5' GGTTAAACCCAGCTTGACCA 3' and 5' GGAGGCAGAGACAGTTGGAG 3'). Experimental animals resulted from crosses between Kdm6a[Flox/+] and Trp53[Flox/+] mice.

## Adenovirus infection of organoids

To generate Trp53[-/-], Kdm6a[-], and Trp53[-/-];Kdm6a[-] organoids, Trp53[Flox/Flox], Kdm6a[Flox], and Trp53[Flox/Flox];Kdm6a[Flox] organoids were infected with Ad5CMVCre-eGFP adenovirus (University of Iowa Viral Vector Core) in vitro. Organoids were trypsinized into single cells. 500,000 cells were pelleted in a 1.5-mL tube and re-suspended in 100 µL of organoid medium supplemented with a final concentration of 5 µg/mL polybrene (TR-1003-G Millipore) with or without virus at a MOI of 50. Cells were incubated at 32°C with shaking at 700 rpm on an Eppendorf ThermoMixer for 1 hr. Cells were then incubated at 37°C without shaking for approximately 2.5 hr. Cells were pelleted and re-suspended in 650 µL Matrigel to make twelve 50 µL Matrigel tabs. Tabs were hardened at 37°C for 15 min and organoid medium was added. To verify recombination of floxed alleles, DNA was isolated from organoid cell pellets using the DNeasy Blood and Tissue Kit (69506 Qiagen, Germany) and genotyped via PCR using the primers listed in Key Resources.

## Expression vector construction

Wild-type human RXRA was expressed from pBABE puro RXRA (plasmid #11441 Addgene). Point mutations were introduced via the QuikChange Site-Directed Mutagenesis protocol (Agilent

Technologies). pBABE puro empty vector was generated by digesting pBABE puro RXRA with EcoRI to remove RXRA and by ligating the plasmid ends together with T4 DNA ligase (M0202S New England BioLabs). Wild-type human PPARG1 was expressed from pCMV6-XL4 PPARG (SC108192 Origene). Point mutations were introduced via the QuikChange Site-Directed Mutagenesis protocol. pCMV6-XL4 empty vector was generated by digesting pCMV6-XL4 PPARG with NotI to remove PPARG and by ligating the plasmid ends together with T4 DNA ligase. Human PPARD was cloned from JMSU-1 epithelial bladder cancer cells. Cells were harvested upon reaching 80–90% confluency and total RNA was isolated using the RNeasy Mini Kit. JMSU-1 cDNA was generated using a First-Strand cDNA Synthesis Kit (27-9261-01 GE Healthcare Life Sciences) with random hexameric primers and 5 µg total RNA. Primers targeting the 5' and 3' UTRs were used to amplify PPARD. PPARD was inserted into the pCMV6-XL4 expression vector using the In-Fusion HD Cloning System (639646 Clontech). Point mutations were introduced via the QuikChange Site-Directed Mutagenesis protocol. See Key Resources for primer sequences.

### Cell lines

JMSU-1, 575A, and UM-UC-3 were obtained from Dr. David Solit (MSKCC). 293 Lenti-X cells were obtained from Clonetech. Cells were cultured in recommended medium supplemented with 10% fetal bovine serum (FBS; FB-11 Omega Scientific), 1% penicillin-streptomycin (15140122 Gibco), and 1x GlutaMAX Supplement (35050061 Gibco). Cells were grown at 37°C in 5% $CO_2$.

### Retroviral infections

Retrovirus was produced by using LipoD293 DNA In Vitro Transfection Reagent (SL100668 SignaGen Laboratories) to transiently transfect 293 Lenti-X cells with 0.45 µg VSVG, 2.27 µg pCL-ampho, and 2.27 µg pBABE puro empty vector, RXRA WT, RXRA S427F, or RXRA S427Y in a 100 mm TC-treated tissue culture dish. Medium was replaced with recommended culturing medium approximately 6 hr post-transfection. Retrovirus was collected 48 hr post-transfection and was filtered through a 0.45-µM filter. Stable cells lines were generated by seeding cells in a 100 mm TC-treated tissue culture dish 18–24 hr before adding virus so that cell confluency was 50–70% when virus was added. The next day, medium was removed and retrovirus diluted 1:4 in the recommended culturing medium and supplemented with a final concentration of 8 µg/mL polybrene (TR-1003-G Millipore) was added to the cells. After approximately 24 hr, the virus-containing medium was replaced with fresh medium. Cells were cultured for an additional 36 hr, at which time puromycin (P8833 Sigma-Aldrich) was added at a predetermined concentration to select for transduced cells. To generate stable organoid lines, organoids were trypsinized into single cells. 500,000 cells were pelleted in a 1.5-mL tube and resuspended in 500 µL of 1:1 organoid medium:retrovirus mixture supplemented with a final concentration of 8 µg/mL polybrene. Cells were incubated at 32°C with shaking at 600 rpm on an Eppendorf ThermoMixer for 1.5 hr. Cells were then incubated at 37°C without shaking for approximately 5 hr. Cells were pelleted and resuspended in enough Matrigel to make three 50 µL Matrigel tabs. Tabs were hardened at 37°C for 15 min and organoid medium was added. 48 hr after infection, puromycin was added at a predetermined concentration to select for transduced cells.

### siRNA knockdown

Cells were seeded in a 12-well cell culture plate 18–24 hr before transfection so that cell confluency was 50–70% at time of transfection. PepMute siRNA Transfection Reagent (SL100566 SignaGen Laboratories) was used to transiently transfect cells with ON-TARGETplus Non-targeting Pool (D-001810-10-20 Dharmacon), ON-TARGETplus Human PPARG siRNA (L-003436-00-0005 Dharmacon), and/or ON-TARGETplus Human PPARD siRNA (L-003435-00-0005 Dharmacon). PepMute/siRNA complex-containing medium was replaced with recommended culturing medium approximately 24 hr post transfection. Cells were collected 72 hr post transfection.

### Western blotting

Cells and organoids were lysed in M-PER (78501 Thermo Scientific) supplemented with 1x Halt Protease Inhibitor Cocktail (87786 ThermoFisher Scientific) and 0.45 M NaCl for 10 min on ice with periodic vortexing. Lysates were cleared by centrifugation at 18,000 x g for 15 min. Protein concentration was determined with Pierce BCA Protein Assay (23224/23228 ThermoFisher Scientific).

Protein lysates were denatured by boiling in 1x Bolt LDS Sample Buffer (B0008 Invitrogen) and 1x NuPAGE Sample Reducing Reagent (NP0004 Invitrogen) for 10 min. Proteins were separated on Bolt 4–12% Bis-Tris Plus Gels (NW04122BOX Invitrogen), transferred onto Immobilon-P Membrane, PVDF, 0.45 µM (IPVH00010 Millipore), and blocked according to antibody specifications. Blots were incubated with primary antibody in blocking solution overnight at 4°C. Blots were washed with 1x TBS plus 0.1% Tween and primary antibodies were detected with HRP-conjugated secondary antibodies. Amersham ECL Prime western blotting detection reagent (RPN2232 GE Healthcare Life Sciences) or Clarity Western ECL Substrate (170–5060 Bio-Rad) was used for chemiluminescence and luminescence was detected with the Bio-Rad ChemiDoc XRS + System. Antibodies and concentration list in Key Resources

## RNA isolation, cDNA synthesis, and RT-qPCR

RNeasy Mini Kit (74106 Qiagen) was used to isolate RNA from organoid and cell pellets. iScript cDNA Synthesis Kit (1708891 Bio-Rad) was used to synthesize cDNA from 0.5 to 1 µg of RNA. RT-qPCR was performed according to package instructions for SsoFast EvaGreen Supermix with low ROX (172–5211 Bio-Rad) on the Applied Biosystems QuantStudio 3 Real-Time PCR system. See Key Resources for primer sequences used.

## Reporter assays

Cells were seeded in a 48-well cell culture plate 18–24 hr before transfection so that cell density was approximately 70% at time of transfection. LipoD293 DNA In Vitro Transfection Reagent (SL100668 SignaGen Laboratories) was used to transiently transfect cells with a reporter plasmid, expression vectors, and a plasmid expressing *Renilla* luciferase. LipoD293/DNA complex-containing medium was replaced with recommended culturing medium supplemented with 1% FBS 16–18 hr post-transfection. Drugs were added to cells 48 hr post transfection. Drug-containing medium was removed 16 hr later and cells were washed with 1x PBS. Luciferase activity was assayed using Dual-Glo Luciferase Assay System (E2940 Promega) and in accordance with the Dual-Glo protocol except that a 1:1 mixture of 1xPBS:Dual-Glo Reagent was added to the cells. Luminescence was measured on the SpectraMax i3 Platform (Molecular Devices). See Key Resources for list of plasmids.

## Growth assays

Organoids were plated at 1000 or 1500 cells per tab and cultured in normal or EGF deplete organoid medium, respectively, for 8–10 days. Drugs or DMSO were added to medium at time of plating. Medium was changed every 4 days post-plating. Cell viability was measured day 1 (baseline measurement) and at indicated days post-plating using CellTiter-Glo (G7571 Promega). Briefly, tabs were homogenized in 1:1 mixture of 1xPBS:CellTiter-Glo reagent and incubated at room temperature for 10 min with shaking. Homogenate was transferred to a white, 96-well plate and luminescence was measured on the SpectraMax i3 Platform. For organoid counting assay, organoids were digested into a single-cell suspension with trypsin and the counted in duplicate using a Bio-rad TC20.

## RNA-seq sample preparation

Cells were seeded in biological triplicate in 6-well tissue culture plates at 300,000 cells (JMSU-1) or 125,000 (575A) cells per well. 48 hr later, 0.1% DMSO or 1 µM pioglitazone (E6910 Sigma-Aldrich) was added to cells. RNA was isolated with the RNeasy Mini Kit (74106 Qiagen) 16–18 hr after addition of drugs. RNA library prep and sequencing were done by the Genome Technology Access Center in the Department of Genetics at Washington University School of Medicine. Briefly, Ribosomal RNA was removed by poly-A selection using Oligo-dT beads. mRNA was then fragmented and reverse transcribed to yield double stranded cDNA using random hexamers. cDNA was blunt ended, had an A base added to the 3' ends, and then had Illumina sequencing adapters ligated to the ends. Ligated fragments were then amplified for 12 cycles using primers incorporating unique index tags. Fragments were sequenced on an Illumina HiSeq-2500 or HiSeq-3000 using single reads extending 50 bases.

## RNA-seq data acquisition, quality control, and processing

RNA-seq reads were aligned to the Ensembl release 76 top-level assembly with STAR version 2.0.4b. Gene counts were derived from the number of uniquely aligned unambiguous reads by Subread:featureCount version 1.4.5. Transcript counts were produced by Sailfish version 0.6.3. Sequencing performance was assessed for total number of aligned reads, total number of uniquely aligned reads, genes and transcripts detected, ribosomal fraction known junction saturation and read distribution over known gene models with RSeQC version 2.3. All gene-level and transcript counts were then imported into the R/Bioconductor package EdgeR and TMM normalization size factors were calculated to adjust for samples for differences in library size. Genes or transcripts not expressed in any sample were excluded from further analysis. The TMM size factors and the matrix of counts were then imported into R/Bioconductor package Limma and weighted likelihoods based on the observed mean-variance relationship of every gene/transcript and sample were then calculated for all samples with the voomWithQualityWeights function. Performance of the samples was assessed with a spearman correlation matrix and multidimensional scaling plots. Gene/transcript performance was assessed with plots of residual standard deviation of every gene to their average log-count with a robustly fitted trend line of the residuals. Generalized linear models were then created to test for gene/transcript level differential expression. Differentially expressed genes and transcripts were then filtered for FDR adjusted p-values less than or equal to 0.05. Go-Elite version 1.2 was used to identify over-represented pathways curated by KEGG.

## ChIP-seq sample preparation

JMSU-1 cells stably expressing RXRA$^{wt}$ or RXRA$^{S427F}$ were seeded in biological triplicate in 150 mm TC-treated culture dish and collected when 70–90% confluency was reached. Crosslinking and ChIP were performed as described for adherent cells in the ENCODE experiment summary for ENCSR000BJW (https://www.encodeproject.org/experiments/ENCSR000BJW/). Chromatin was sonicated using the Diagenode Pico Bioruptor. DNA-protein complexes were precipitated with antibodies against RXRA (PP-K8508-00 R and D Systems) and H3K27Ac (ab4729 Abcam). Ovation Ultralow System V2 (0344–32 NuGen) was used to generate the sequencing library. The library was sequenced by the Genome Technology Access Center in the Department of Genetics at Washington University School of Medicine.

## ChIP-seq data analysis

Reads were aligned to hg19 using NovoAlign version 3.04.06 with the command novoalign -r None -l 30 -e 100 -i 230 140 –H. For the RXRA and H3K27ac IPs, peaks were called for each phenotype using the IDR (*Li et al., 2011*) pipeline described here (*Kundaje, 2012*) with MACS version 2.1.1 (*Zhang et al., 2008*) as the peak caller. RAE regions were defined for each phenotype by running bedtools intersect on the H3K27ac and RXRA IDR peak lists. Differential binding analysis was performed on the H3K27ac signal using DiffBind (*Ross-Innes et al., 2012*) comparing mutant to wild-type with the IDR peak lists as input and an FDR cutoff of 0.05. Motif analysis of the RAE regions was performed using HOMER version 4.9 (*Heinz et al., 2010*) with the command findMotifsGenome.pl –h –bg <Background> <Peaklist> hg19. Peaklist was the RAE regions found by DiffBind to have higher H3K27ac binding in mutant. Background was the list of all RAE regions minus the ones found in peaklist.

## Preparation of structures for molecular dynamics simulations

The published structure of complexed RXRA/PPARG LBDs bound to agonist (PDB: 1FM6) (*Gampe et al., 2000*) was used as the starting conformation for simulations. To reduce subsequent computational complexity and enable drift towards an inactive state during simulations, the respective agonist structures binding RXRA and PPARG were removed. The small peads were aligned to hg19 usptide nuclear receptor coactivator 1 (NCOA1) bound to both RXRA and PPARG AF2 domains was also removed with the same purpose. The Chimera molecular visualization tool (*Pettersen et al., 2004*) was used to mutate RXRA serine 427 to phenylalanine (RXRA S427F), creating wild-type and mutant structures for subsequent simulations.

## Molecular dynamics simulations

The protocol described by Hart et. al. was used for molecular dynamics simulations (*Hart et al., 2016*). Briefly, simulations were performed using the molecular dynamics package GROMACS 5.1.3 (*Abraham et al., 2015*). The Amber03 force field (*Duan et al., 2003*) was used and hydrogen atoms in each structure were replaced with virtual sites. The protein structure was solvated within a dodecahedron whose border was at least 10 Å away from the protein in all directions. Chlorine counterions were added to neutralize the system's overall positive charge. To ensure the system contains no steric clashes or inappropriate geometry, energy minimization was used to relax the system below a 1000 kJ/mol/nm threshold. To equilibrate solvent and ions around the structure, the system underwent position-restrained molecular dynamics simulation for one nanosecond using a step size of 4 fs. After relaxation and equilibration, each system was subjected to long-timescale molecular simulations. Simulations took place using the NPT ensemble at 1 bar and 300 K. Parrinello-Rahman pressure coupling (*Parrinello and Rahman, 1981*) and the V-rescale thermostat (*Bussi et al., 2007*) were used during simulations. The LINCS method (*Hess, 2008*) constrained hydrogen bonds and allowed the use of virtual sites. The cutoffs for electrostatic and van der Waals interactions was 9 Å. Periodic boundary conditions were applied during simulation and the particle-mesh Ewald summation reconstituted any long-distance electrostatic interactions. Each simulation lasted 100 nanoseconds with conformations stored every 10 picoseconds. Ten simulations were run to produce an aggregate 1 microsecond of simulation for both wild-type and mutant conditions.

## Markov state modeling

MSMBuilder 2.8 (*Beauchamp et al., 2011*) was used to analyze conformations adopted during molecular simulation. Wild-type and mutant structures were clustered in overlapping state space using the hybrid k-centers/k-medoids technique. Clustering was based on the RMSD of between backbone-heavy atoms from PPARG 445–477 and RXRA 425–429. The inter-cluster distance cutoff was 1 Å and 50 iterations were performed to refine cluster assignment. A microstate Markov state model (MSM) was constructed using the Transpose method for symmetric counts matrix estimation and without applying an ergodic trim. The 5% most frequently occupied microstates during wild-type and mutant conformations were selected. Distance between alpha carbons of RXRA 427 and PPARG 477 was calculated for comparison between microstate structures from wild-type and mutant simulations. The inter-residue distances were normalized to the respective distance in the agonist-bound crystal structure. The Chimera molecular visualization tool was used to generate representative figures.

## Statistics

All statistical tests are as indicated in figure legends. Analysis was done in GraphPad Prism with the exception of pearson coefficients which were calculated using Microsoft Excel. All statistical comparisons are two-tailed. *$p<0.05$, **$p<0.01$, ***$p<0.001$, ****$p<0.0001$.

## Acknowledgements

We are grateful to Harrison Gabel (WUSM) for helpful discussions and technical advice for ChIP-seq experiments, Sukrit Singh (WUSM), Maxwell Zimmerman (WUSM), and Justin Porter (WUSM) for technical guidance for molecular dynamics studies, and Gopa Iyer (MSKCC) and Ricardo Ramirez (MSKCC) for helpful discussions and guidance on bladder cancer cell line selection.

## Additional information

### Funding

| Funder | Grant reference number | Author |
|---|---|---|
| Damon Runyon Cancer Research Foundation | Clinical Investigator | Vivek K Arora |
| Cancer Research Foundation | Young Investigator | Vivek K Arora |
| National Cancer Institute | T32 CA113275 | Angela M Halstead |

| National Center for Advancing Translational Sciences | UL1TR000448 | Vivek K Arora |
| National Cancer Institute | P30 CA91842 | Andrew Schriefer |
| National Institute of Diabetes and Digestive and Kidney Diseases | U54DK104279 | Chiraag D Kapadia |

The funders had no role in study design, data collection and interpretation, or the decision to submit the work for publication.

### Author contributions

Angela M Halstead, Conceptualization, Data curation, Formal analysis, Funding acquisition, Investigation, Methodology, Writing—original draft; Chiraag D Kapadia, Conceptualization, Data curation, Formal analysis, Validation, Investigation, Visualization, Methodology, Writing—original draft; Jennifer Bolzenius, Data curation, Formal analysis, Validation, Investigation, Methodology; Clarence E Chu, Data curation, Formal analysis, Investigation, Visualization, Methodology; Andrew Schriefer, Data curation, Formal analysis, Methodology; Lukas D Wartman, Resources, Methodology, Writing—review and editing; Gregory R Bowman, Conceptualization, Resources, Formal analysis, Supervision, Visualization, Methodology, Writing—original draft; Vivek K Arora, Conceptualization, Data curation, Formal analysis, Supervision, Funding acquisition, Investigation, Visualization, Methodology, Writing—original draft

### Author ORCIDs

Lukas D Wartman [iD] http://orcid.org/0000-0002-5499-8465
Vivek K Arora [iD] http://orcid.org/0000-0003-1694-9109

### Ethics

Animal experimentation: All mouse experiments were performed in accordance with institutional guidelines and current NIH policies and were approved by the Washington University School of Medicine Institutional Animal Care and Use Committee protocol #20140186.

### Decision letter and Author response

Decision letter https://doi.org/10.7554/eLife.30862.018
Author response https://doi.org/10.7554/eLife.30862.019

## Additional files

### Supplementary files

• Transparent reporting form
DOI: https://doi.org/10.7554/eLife.30862.014

### Major datasets

The following previously published dataset was used:

| Author(s) | Year | Dataset title | Dataset URL | Database, license, and accessibility information |
| --- | --- | --- | --- | --- |
| Cerami E, Gao J, Dogrusoz U, Gross BE, Sumer SO, Aksoy BA, Jacobsen A, Byrne CJ, Heuer ML, Larsson E, Antipin Y, Reva B, Goldberg AP, Sander C, Schultz N | 2012 | cBio Cancer Genomics Portal | https://www.ncbi.nlm.nih.gov/geo/query/acc.cgi?acc=GSE107783 | Publicly available at the NCBI Gene Expression Omnibus (accession no. GSE107783) |

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
