## [Decision Letter]

Thank you for submitting your article "Hot-spot Mutations in *RXRA* Implicate Peroxisome Proliferator-Activated Receptors as Bladder Cancer Drivers" for consideration by *eLife*. Your article has been favorably evaluated by Kevin Struhl (Senior Editor) and three reviewers, one of whom, Ross L Levine (Reviewer #1), is a member of our Board of Reviewing Editors. The following individuals involved in review of your submission have agreed to reveal their identity: Kendall W Nettles (Reviewer #2); Jonathan Kurie (Reviewer #3).

The reviewers have discussed the reviews with one another and the Reviewing Editor has drafted this decision to help you prepare a revised submission.

Summary:

The work here showing a role for *RXRA* mutations in bladder cancer pathogenesis is of interest, however some additional studies and revisions to the paper are needed to further underscore these results and to place them in their proper context. In particular, although *RXRA* has been implicated in bladder cancer pathogenesis based on studies of overexpression, the novelty here is showing how these mutants contribute to oncogenic transformation.

Essential revisions:

1) The authors should revise the title and conclusions to avoid overstatement, because the oncogenic role of PPAR signaling in bladder cancer has been widely studied. It has been reported that PPARG is amplified in 17% bladder cancer, while *RXRA* mutation only happens in about 5% patients (and maybe there is some overlap). However, the mechanism that *RXRA* mutation functions by activating the PPAR pathway is interesting and should be highlighted and discussed given this important, novel aspect.

2) The authors found that *RXRA* mutant can activate PPARG independent of PPAR ligand binding using molecular simulations with the crystal structures. This finding is interesting. It will be valuable if the authors can go further at this point and provide some ideas on how to block the active site of PPAR based on these findings in the Discussion section.

3) It would be critical to show that the key target genes altered by mutant *RXRA* in their model systems are similarly altered in primary samples with mutant *RXRA*.

4) Can the authors distinguish between WT and mutant *RXRA* binding sites in the same cell (e.g. with 2 different tagged constructs), given cancer cells will have WT and mutant protein?

5) For the organoid culture, and luciferase data, biological replicates should be from experiments performed at different times, with technical replicates deriving from cells grown in different wells. It is essential that this has been done and that this information is clearly stated in the manuscript. Figure 4 states that data from organoids is from triplicate wells (assayed in duplicate). This is not clear. The experimental design should be something more like: "data is the mean + SD of three experiments, each of which was assayed in triplicate wells on separate days".

---

## [Author Response]

Essential revisions:1) The authors should revise the title and conclusions to avoid overstatement, because the oncogenic role of PPAR signaling in bladder cancer has been widely studied. It has been reported that PPARG is amplified in 17% bladder cancer, while RXRA mutation only happens in about 5% patients (and maybe there is some overlap). However, the mechanism that RXRA mutation functions by activating the PPAR pathway is interesting and should be highlighted and discussed given this important, novel aspect.

We have now changed the title to “Bladder Cancer Associated Mutations in *RXRA* Activate Peroxisome Proliferator-Activated Receptors to Drive Urothelial Proliferation” which better summarizes the main conclusions of the paper. We agree that the characterization of the *RXRA* mutation is the main contribution of the paper. We acknowledge there are several papers that have studied PPARG in bladder cancer, but its role as a driver pathway in bladder cancer is not something (in our view) yet firmly established or accepted in the bladder cancer field. Thus, our finding that PPAR pathway activation drives urothelial proliferation is, we believe, an important secondary contribution of the paper. We have made changes in the Abstract and Discussion to better articulate the relative importance of these views.

2) The authors found that RXRA mutant can activate PPARG independent of PPAR ligand binding using molecular simulations with the crystal structures. This finding is interesting. It will be valuable if the authors can go further at this point and provide some ideas on how to block the active site of PPAR based on these findings in the Discussion section.

We have extended our previous Discussion sharing more of our thoughts about strategies that might be effective for inhibiting PPAR activity in the context of the *RXRA* mutations.

3) It would be critical to show that the key target genes altered by mutant RXRA in their model systems are similarly altered in primary samples with mutant RXRA.

· The ‘key target genes’ analyzed by qPCR in the manuscript include *PLIN2* and *FABP3/4. PLIN2* is up-regulated in the human samples with *RXRA* hot-spot mutations and was utilized in all of our qPCR analysis for this reason. This rationale has now been indicated in the manuscript. In Supplementary Figure S2.6 (RXRA hotspot mutations) from the TCGA’s Nature 2014 paper (Nature 507, 315–322, 2014) a volcano plot shows altered gene expression in *RXRA* hot-spot mutant clinical sample with *PLIN2* located near the top right of the plot. The FABPs did not score individually as up-regulated in mutant cases, but are canonical PPAR target genes. These genes showed particularly strong expression signal changes associated with the mutation in cell lines and organoids, and therefore provided a broad signal range to assess the impact of siRNA knock-down or drug treatments. Notably, there are very few individual genes clearly altered in the mutant cases versus wild-type, and the statistical significance shown in the plot is modest for discovery analysis. We presume this is in part because gene expression values of individual genes are inherently ‘noisy’ in human tissues and the *RXRA* hot-spot mutation is still limited to a small number of patient samples. Gene Pathway analysis we believe helps overcome some of the ‘noise’ associated with gene expression analysis and is a particularly useful tool when assessing changes in transcription factor activity. As we indicated in the manuscript, *RXRA* mutation is associated with PPAR signaling pathway up-regulation in *human tissues*, and a causal role of *RXRA* mutation for this pathway up-regulation was confirmed in two cell lines in our study.

4) Can the authors distinguish between WT and mutant RXRA binding sites in the same cell (e.g. with 2 different tagged constructs), given cancer cells will have WT and mutant protein?

We agree with the reviewers that such an approach could provide an elegant demonstration of differences in WT and mutant *RXRA* DNA binding and enhancer activation. However, such an approach would also introduce two new potentially confounding variables: 1) different tags which might differentially alter protein function, and 2) the need to use two different antibodies for ChIP with each introducing its own bias with regard to DNA binding site identification and background binding. While with appropriate controls these issues could be surmountable, we feel that adequate validation of the approach would take several months and therefore we cannot include it at this time. We feel that the approach we used, introduction of not tagged wild-type or mutant *RXRA* independently into cell lines and then performing ChIP for *RXRA* and H3K27ac, is adequate to conclude that hyper-activation of enhancer/promoters driven by the *RXRA* hot-spot mutation occurs preferentially at canonical RXR/PPAR response elements.

5) For the organoid culture, and luciferase data, biological replicates should be from experiments performed at different times, with technical replicates deriving from cells grown in different wells. It is essential that this has been done and that this information is clearly stated in the manuscript. Figure 4 states that data from organoids is from triplicate wells (assayed in duplicate). This is not clear. The experimental design should be something more like: "data is the mean + SD of three experiments, each of which was assayed in triplicate wells on separate days".

We thank the reviewers for highlighting this lack of clarity and have removed the terms ‘biological’ and ‘technical’ replicate as they are ambiguous. In our previous submission, plotted data for the organoid growth and luciferase data came from one experiment in which cells were plated into three different wells (triplicate) and then two equal volume samples were taken from each well for the endpoint bioluminescence assay (duplicate). Data shown was considered representative of multiple experiments. In response to this comment, we have reformatted many of the data panels such that all the organoid and luciferase experiments include data from three distinct experiments performed on three different days. Since this affects many panels, we are uploading an appendix showing the “old” figure panel next to the “new” figure panel to facilitate review of all affected panels. None of the conclusions of the manuscript are affected and the data look quite similar. The methodology for plotting and analyzing data collected from experiments executed on different days is summarized below. We also point out minor differences resulting from these data analysis.

1) For panels 1F, 2D, 4C, 4F, 5B, 5C, 5D, and 5E, we have now plotted the mean bioluminescence signal from three identical and independent experiments performed on different days (each performed using 3 replicate cell wells, with 2 independent bioluminescence measurements from each cell well.) For these assays, there is inherently some variation in *absolute* raw bioluminescence signal from day to day, presumably due to a range of factors (e.g. temperature, reagent batches, variations in cell density, etc.). The *relative* bioluminescence signal between conditions is consistent when comparing experiments performed on different days. Since the signal ratios between biological conditions is of greater relevance then the bioluminescence absolute value, we have used a (ratio) paired *t*-test (instead of an unpaired *t-*test) for the statistical comparisons of integrated data from multiple experiments. All data from an experiment performed on the same day are “paired”.

2) For Figure 3, as we did in the original submission, the data for each PPAR allele is plotted as a ratio normalized to the *RXRA* WT DMSO condition for that PPAR allele. This approach provides a more intuitive visualization of the effect of PPAR mutations on inducibility by either agonist or *RXRA* S427F. Since with this approach the data for each experimental replicate is already expressed as a normalized ratio to another experimental condition assayed on the same day, comparison is with an unpaired *t* test. The plotted data now represents the mean ratios from three independent experiments performed on different days, each using cell wells plated in triplicate. (Statistical comparisons were not included in the previous submission, but have been added for completeness.)

For Figure 4 and Figure 4—figure supplement 1 we have also now plotted the mean bioluminescence signal from three independent experiments performed on different days (each performed using 3 replicate cell wells, with 2 independent bioluminescence measurements from each cell well.) As before, these assays were performed using three independently derived organoids lines for each genotype. The statistical comparisons are between the mean values from the three independently derived organoids after each drug treatment. Thus, the variation in bioluminescence signal of experiments performed on different days does not apply here and an unpaired t-test analysis was used.

In the previous submission, a significant and consistent impact of GW0742 on growth was limited to organoids deficient in the tumor suppressors Kdm6a and Trp53. As we had pointed out, the data with Trp53 knock-out alone trended toward an effect but it was not statistically significant. By incorporating data from multiple experiments (presumably due to increased statistical power), GW0742 also drives significant growth in organoids with Trp53 deletion alone, although still not with the same apparent consistency across the independently derived lines compared to the dual Kdm6a/Trp53 knock-out organoids. Nonetheless, we have changed language to indicate that Trp53 alone is sufficient for GW0742 to drive growth factor independent growth. We again show that GW0742 fails to drive growth factor independent growth in wild-type organoids. Our conclusion that “PPARD activation induces growth factor independent growth of urothelial organoids in the context of tumor suppressor loss” is not changed.

Figure 4 and Figure 5 show qPCR data expressed relative to the housekeeping gene actin from one experiment. The entire experiment was performed three times (i.e. on 3 different days.) Because again of inherent noise in the absolute signal value with qPCR data from different experiments, we have kept data from each experiment separate, but have included all data in 2 new supplemental panels (Figure 4—figure supplement 1 and Figure 5—figure supplement 1.) Statistical comparison of the aggregated data is by two-way ANOVA (Repeated Measures, GraphPad Prism).

In Figure 5, both PPARD inhibitors (GSK0660 and ST247) are again shown to significantly impair growth and transcriptional activity driven by RXRA S427F. However, with integrated analysis, one inhibitor does not clearly perform better than the other, therefore a suggestion we had made that ST247 may be more efficacious has been removed.